# Uncertainty-Guided Exploration and Stable Planning for Sparse-Reward Manipulation from Limited Demonstrations

**Haowen Sun** [1]  **Liqi Huang** [1]  **Mingyang Li** [1]  **Sihua Ren** [1]  **Xinzhe Chen** [1]  **Chengzhong Ma** [1]  **Zeyang Liu** [1]  **Xingyu Chen** [1]  **Xuguang Lan** [1]

## Abstract

Reinforcement learning from demonstrations (RLfD) offers a promising method for robotic manipulation with sparse rewards. However, limited demonstrations often cause agents to encounter out-of-distribution states where world models produce poor predictions. In multi-stage tasks, jointly optimizing a learned reward function and policy introduces a moving target problem, and the resulting non-stationarity intensifies the impact of uncertainty on policy learning. In this work, we propose QUEST, a model-based RL framework that adaptively switches between exploration and exploitation guided by uncertainty to achieve stable and efficient learning. Specifically, our approach employs intrinsic rewards to encourage exploration, leverages ensemble dynamics for uncertainty-guided planning, and introduces a hybrid sampling strategy to prioritize rare successful stage transitions. We evaluate QUEST on challenging sparse-reward manipulation tasks with limited expert demonstrations. Results show that QUEST outperforms state-of-the-art methods by 17% on average, with gains increasing to 60% on difficult tasks. We further demonstrate successful zero-shot sim-to-real transfer on five real-world tasks. Project website: `https://quest-official.github.io/QUEST/`.

## 1. Introduction

Reinforcement learning from demonstrations (RLfD) offers a promising paradigm for addressing exploration challenges in sparse-reward robotic manipulation tasks by leveraging expert guidance (Vecerik et al., 2017; Rengarajan et al., 2022; Hu et al., 2025). Specifically, model-based reinforcement learning (MBRL) utilizes a world model to predict future outcomes and plan optimal actions, improving learning efficiency (Hansen et al., 2022b; Hafner et al., 2019; Sekar et al., 2020). However, these methods typically require extensive demonstrations to ensure sufficient coverage of the state-action space (Trott et al., 2019; Wu et al., 2021; Memarian et al., 2021; Escontrela et al., 2022). This requirement becomes prohibitive as task complexity increases, particularly in multi-stage tasks where acquiring high-quality expert data is costly. With limited expert data, the agent frequently encounters out-of-distribution (OOD) states, where the world model produces poor predictions, leading to flawed planning and undermining policy learning.

To mitigate data requirements, prior works leverage data generation to augment the replay buffer (Hansen et al., 2022a; Lancaster et al., 2024; Zhan et al., 2022) or employ imitation learning to construct dense reward functions (Hu et al., 2023; Shi et al., 2022), facilitating exploration in subsequent RL. Although these approaches effectively expand the state-action space, the distribution gap between the RL policy and expert demonstrations remains a challenge. When the agent enhances exploration to improve policy performance, it inevitably encounters states beyond the expert data distribution (Ross et al., 2011). A promising direction to overcome these limitations is to jointly optimize a learned reward function alongside the policy, enabling both components to improve collaboratively (Zhao et al., 2025; Escoriza et al., 2025). However, this joint optimization introduces a moving target problem for policy learning, causing instability in policy optimization.

We find that the moving target problem introduced by joint optimization causes non-stationarity, intensifying the impact of uncertainty on policy learning in robotic multi-stage tasks. This uncertainty includes both aleatoric and epistemic components (Kendall & Gal, 2017). Aleatoric uncertainty arises from complex object interactions and domain randomization, while epistemic uncertainty stems from insufficient learning of the policy and world model. Both components

---

[1]National Key Laboratory of Human-Machine Hybrid Augmented Intelligence, Institute of Artificial Intelligence and Robotics, Xi'an Jiaotong University, Xi'an, China. Correspondence to: Xingyu Chen <chenxingyu_1990@xjtu.edu.cn>, Xuguang Lan <xglan@mail.xjtu.edu.cn>.

*Proceedings of the $43^{rd}$ International Conference on Machine Learning*, Seoul, South Korea. PMLR 306, 2026. Copyright 2026 by the author(s).

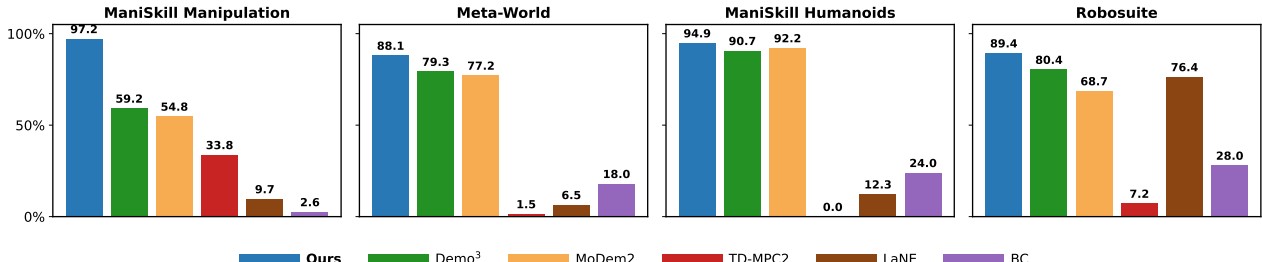

*Figure 1.* **Summary of results**. Final success rate (%) achieved by our method and baselines, averaged across 4 domains (avg. 5 seeds). Our method outperforms strong baselines in sparse-reward visual manipulation tasks using only 10 demonstrations.

affect planning. Without sufficient supervision, increased exploration under such uncertainty may drive the agent toward task-irrelevant regions, destabilizing reward learning and impairing world model learning. Therefore, the policy should dynamically balance exploration and exploitation according to environmental conditions and learning progress.

In this paper, we propose Quantifying Uncertainty to Enable Effective Exploration and Stable Planning (QUEST), a method that adaptively switches between exploration and exploitation guided by uncertainty to achieve stable and efficient learning. The core insight is that enhancing exploration mitigates uncertainty, while simultaneously leveraging the quantified uncertainty to adaptively switch conservative planning strategies for stable learning. Specifically, we employ Random Network Distillation (RND) (Burda et al., 2018) to compute intrinsic rewards exclusively for Q-function updates, enabling the critic to accurately capture environmental stochasticity. Furthermore, we introduce adaptive uncertainty-guided planning that leverages ensemble dynamics for uncertainty quantification. Additionally, a hybrid sampling strategy accelerates policy learning by prioritizing rare successful stage transitions.

We evaluate QUEST on a range of challenging sparse-reward manipulation tasks from Meta-World (Yu et al., 2020a), Robosuite (Zhu et al., 2020), and ManiSkill3 (Tao et al., 2024) in long-horizon multi-stage manipulation with limited expert demonstrations. Our results (see Figure 1) demonstrate that QUEST outperforms state-of-the-art methods by an average of 17%, with performance gains increasing to 60% on more complex manipulation tasks. Furthermore, we validate QUEST on real-world robotic manipulation tasks, demonstrating successful zero-shot sim-to-real transfer across five distinct tasks.

## 2. Preliminaries

**Problem formulation.** We formulate the problem of learning control policies for multi-stage manipulation tasks as an infinite-horizon Markov Decision Process (MDP) (Bellman, 1957) defined by the tuple $(\mathcal{S}, \mathcal{A}, \mathcal{P}, \mathcal{R}, \gamma)$, where $\mathcal{S}$ and $\mathcal{A}$

denote the state and action spaces, $\mathcal{P}$ is the unknown transition function, $\mathcal{R}$ is a sparse reward function, and $\gamma \in [0, 1)$ is the discount factor. Our goal is to learn a policy $\pi : \mathcal{S} \rightarrow \mathcal{A}$ that maximizes the objective $J_{\mathcal{M}}(\pi) = \mathbb{E}_\pi[\sum_{t=0}^\infty \gamma^t r_t]$, where $r_t = \mathcal{R}(s_t, \pi(s_t))$. This objective correlates with the state-value function $V_{\mathcal{M}}^\pi(s) = \mathbb{E}_\pi[\sum_{l=0}^\infty \gamma^l r_{t+l} \mid s_t = s]$.

**RL with dense reward learning.** To address the challenge of sparse rewards, we jointly learn a dense reward function alongside the RL policy. We define the state space $\mathcal{S}$ as multi-modal observations $s$, comprising RGB images and proprioceptive states. Adapting the Demo[3] framework (Escoriza et al., 2025), we model the reward structure using stage-specific discriminators. Specifically, a discriminator $\delta_k$ is instantiated for each stage $k$, and its output augments the sparse environmental reward $r_t$ to produce a dense reward:

$$\hat{r}_t = r_t + \beta \cdot \tanh(\delta_k(s_t)), \tag{1}$$

where $\beta$ is a scaling hyperparameter and $s_t$ is the observation at timestep $t$.

We treat $r_t \in \{0, 1, \ldots, K\}$ as the stage indicator at time $t$. Each $\delta_k$ is trained as a binary classifier on samples with $r_t = k$, predicting whether the trajectory eventually reaches a stage beyond $k$. To this end, we define the maximum stage label $\bar{k}_t = \max_{t' \geq t} r_{t'}$, which records the highest stage the trajectory reaches from time $t$ onward, with $\bar{k}_t > k$ signaling that the sub-trajectory eventually progresses beyond stage $k$. The total discriminator loss is the per-stage Binary Cross Entropy loss averaged across stages:

$$\mathcal{L}_{\text{disc}} = \frac{1}{K} \sum_{k=1}^K \mathbb{E}_{(s_t, r_t=k, \bar{k}_t) \sim \mathcal{B}} \left[ \text{BCE}\left(\mathbf{1}_{\bar{k}_t > k}, \delta_k(s_t)\right) \right]. \tag{2}$$

**TD-MPC2.** TD-MPC2 (Hansen et al., 2023) is a model-based RL algorithm that learns a latent-space world model and a terminal value function via temporal difference (TD) learning, and performs planning via Model Predictive Path Integral (MPPI) (Williams et al., 2015) control. The world model consists of an encoder $h_\theta$ that maps a high-dimensional observation $s$ into a latent state $z = h_\theta(s) \in \mathcal{Z}$,

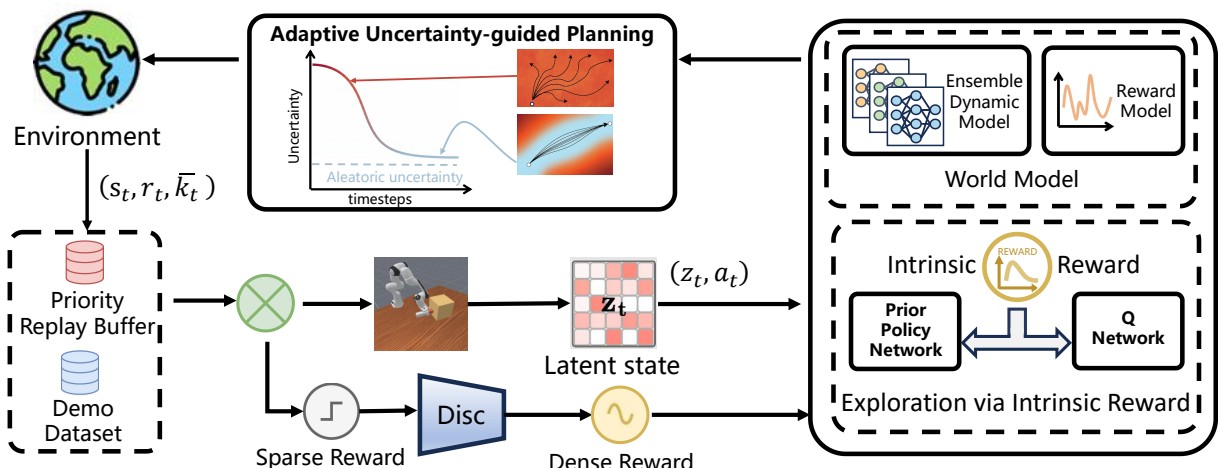

*Figure 2.* **Method overview**. QUEST jointly trains a world model, policy, and dense reward function from limited demonstrations, with a unified uncertainty estimate that orchestrates the transition from exploration in early training to conservative planning later. The environment provides observations $s_t$ and sparse rewards $r_t$. A hybrid sampling strategy draws each batch from a prioritized replay buffer and an expert demonstration buffer to highlight rare successful stage transitions. The encoder $h_\theta$ maps observations to latent states $z_t$ for the world model. Stage-specific discriminators $\delta_k$ convert sparse rewards into dense signals $\hat{r}_t$. During planning, ensemble disagreement $\tilde{\mathcal{U}}$ quantifies uncertainty and controls the trade-off between exploration and exploitation via the adaptively scheduled coefficient $\alpha$. RND intrinsic rewards $r^{\text{intr}}$ are added exclusively to the Q-function TD target, leaving the world-model and reward-predictor losses unaffected.

a dynamics model $d_\theta$ that predicts state transitions as $z' = d_\theta(z, a)$, and a reward predictor $R_\theta(z, a)$. In addition, TD-MPC2 learns a state-action value function $Q_\theta(z, a)$ and a policy prior $a \sim \pi_\theta(z)$. All components except the policy are jointly optimized by minimizing:

$$\mathcal{L}(\theta) = \sum_{i=t}^{t+H} \lambda^{i-t} \Big[ \mathcal{L}_Q(\theta) + \mathcal{L}_R(\theta) + \mathcal{L}_h(\theta) \Big], \quad (3)$$

where $\lambda$ is a temporal weighting factor. In our framework, $\mathcal{L}_R = \text{CE}(R_\theta(z_t, a_t), \hat{r}_t)$ follows TD-MPC2 with the sparse reward $r_t$ replaced by the learned dense reward $\hat{r}_t$. The policy prior $\pi_\theta(z)$ is separately optimized via a loss $\mathcal{L}_\pi$ that maximizes the Q-value with entropy regularization (Haarnoja et al., 2018). During planning, MPPI selects actions and evaluates trajectories using the world model. The estimated value of an action sequence is computed as:

$$\hat{Q} = \gamma^H Q_\theta(z_{t+H}, a_{t+H}) + \sum_{h=0}^{H-1} \gamma^h R_\theta(z_{t+h}, a_{t+h}), \quad (4)$$

where $z_{t+h+1} = d_\theta(z_{t+h}, a_{t+h})$ is the latent state predicted by the dynamics model, $\gamma$ is the discount factor, and $H$ is the planning horizon. The receding-horizon execution of MPPI yields a closed-loop policy $\pi_{\text{p}}$, which is evaluated by maximizing the expected cumulative return as formulated by the objective $J_\mathcal{M}(\pi_{\text{p}})$ earlier.

## 3. Theoretical Analysis of Uncertainty

Multi-stage manipulation tasks involve frequent contact interactions between objects, making uncertainty a central

concern for state transitions. Stochastic contact dynamics, injected noise, and domain randomization over environment parameters all amplify this uncertainty. Formally, given the same state-action pair $(s, a)$, the successor state $s'$ exhibits substantial variability, as it samples from the transition distribution $\mathcal{P}(s' \mid s, a)$. Additionally, the jointly trained reward function introduces a moving target problem, which brings non-stationarity since the stage discriminator $\delta_k$ in the shaped reward $\hat{r}_t$ is continuously updated during training. Moreover, model epistemic uncertainty arises from unreliable predictions of the world model in insufficiently explored regions of the state space, and this uncertainty further propagates through the planning process via the world model and value function.

**Definition 3.1** (Planning Uncertainty). Given a latent state $z \in \mathcal{Z}$ and an action $a \in \mathcal{A}$, we decompose the total uncertainty in model-based planning into three components:

$$\mathcal{U}(z, a) \triangleq \mathcal{U}^{\text{trans}}(z, a) + \mathcal{U}^{\text{rew}}(z, a) + \mathcal{U}^{\text{model}}(z, a). \quad (5)$$

$\mathcal{U}^{\text{trans}}(z, a)$ is the irreducible transition aleatoric uncertainty, capturing the inherent randomness of $\mathcal{P}(s' \mid s, a)$ from stochastic contact dynamics, injected noise, and domain randomization. $\mathcal{U}^{\text{rew}}(z, a)$ captures the reward non-stationarity from joint training, since the continuously updated discriminator $\delta_k$ assigns different reward signals to the same $(z, a)$ pair across stages. $\mathcal{U}^{\text{model}}(z, a)$ captures the model epistemic uncertainty from unreliable predictions of the learned dynamics and value function in data-sparse regions.

In practice, our ensemble disagreement operates entirely in

latent space, where these three components manifest jointly and are difficult to separate reliably. We therefore treat ensemble disagreement as a single unified uncertainty estimate, rather than estimating each component individually. Based on Definition 3.1, we instantiate a computable planning uncertainty proxy $\tilde{\mathcal{U}}(z, a)$ and define the uncertainty-penalized reward:

$$\tilde{R}(z,a) = R_\theta(z,a) + \alpha\tilde{\mathcal{U}}(z,a) \tag{6}$$

where $\alpha < 0$ corresponds to uncertainty penalty (conservative planning) and $\alpha > 0$ corresponds to exploration bonus. This induces an uncertainty-penalized model MDP $\tilde{\mathcal{M}} = (\mathcal{Z}, \mathcal{A}, d_\theta, \tilde{R}, \gamma)$, and we denote by $J_{\tilde{\mathcal{M}}}(\pi_p)$ the expected discounted return of a policy $\pi_p$ under $\tilde{\mathcal{M}}$.

# 4. Method

In this work, we address the challenge of adaptively switching between exploration and exploitation under uncertainty in multi-stage manipulation tasks with sparse rewards and limited demonstrations. As illustrated in Figure 2, we propose QUEST, an MBRL method that leverages ensemble dynamics to quantify uncertainty and adaptively adjust the strategy between exploration and conservative planning.

## 4.1. Exploration via Intrinsic Reward

In sparse-reward tasks with limited demonstrations, the agent needs to enhance exploration to reduce uncertainty in unseen states and enable the world model to capture transition distributions across stochastic environments. Intrinsic motivation provides an effective mechanism for exploration by rewarding visits to novel states. However, directly adding intrinsic rewards to the learned reward function affects both the world model and policy optimization, causing the planner to double-count intrinsic rewards and the world model to deviate from the true environment.

Therefore, we separate intrinsic rewards from world model learning by applying them exclusively to the TD loss, which only affects the Q-function and policy prior while keeping the world model aligned with the real environment. In particular, we employ Random Network Distillation (RND) to compute intrinsic rewards, where a predictor network is trained to match the output of a fixed randomly initialized target network. The prediction error serves as an exploration bonus that is high for novel states and diminishes as states become familiar. The Q-function is updated by minimizing the following TD loss with intrinsic rewards:

$$\mathcal{L}_Q = \mathbb{E}_\mathcal{B}\left[\left(Q_\theta(z,a) - (\hat{r} + r^{\text{intr}} + \gamma Q_{\theta'}(z',a'))\right)^2\right] \tag{7}$$

where $r^{\text{intr}} = \frac{\|\hat{f}(z') - f(z')\|^2}{\sigma_{\text{intr}}}$ denotes the normalized RND intrinsic reward computed as the squared prediction error between the predictor network $\hat{f}$ and the fixed target network

**Algorithm 1** QUEST

---

**Require:** Demonstration dataset $\mathcal{D}$, number of stages $K$, ensemble size $N_e$, planning horizon $H$
1: **Initialize:** encoder $h_\theta$, ensemble dynamics $\{d_{\theta_{n_e}}\}_{n_e=1}^{N_e}$, policy prior $\pi_\theta$, discriminators $\{\delta_k\}_{k=1}^{K}$, RND networks $(\hat{f}, f)$, replay buffer $\mathcal{B} \leftarrow \{\emptyset\}$
2: **Rollout:**
3: **for** each environment step **do**
4:    Initialize trajectory: $\tau \leftarrow \{\emptyset\}$
5:    Encode observation: $z_t \leftarrow h_\theta(s_t)$
6:    Ensemble rollout per candidate action $a$: $z_{t+1}'^{(n_e)} \leftarrow d_{\theta_{n_e}}(z_t, a)$
7:    $a_t \leftarrow \text{MPPI}(\{z_{t+1}'^{(n_e)}\}, \alpha, \tilde{\mathcal{U}}_n)$
8:    $(s_{t+1}, r_t) \leftarrow \text{Env}(a_t)$
9:    $\tau \leftarrow \tau \cup (s_t, a_t, s_{t+1}, r_t)$
10:   **if** episode done **then**
11:      Reset environment
12:      Assign stage labels: $\{\bar{k}_t\}_{t=0}^{T}$
13:      Store trajectory: $\mathcal{B} \leftarrow \mathcal{B} \cup (\tau, \{\bar{k}_t\}_{t=0}^{T})$
14:   **end if**
15: **end for**
16: **Update:**
17: **for** each update step **do**
18:   Sample: $\{(s_t, s_{t+1}, a_t, r_t, \bar{k}_t)\}_{t=t_0}^{t_0+H} \sim (\mathcal{B} \cup \mathcal{D})$
19:   Predict dense reward $\{\hat{r}_t\}_{t=t_0}^{t_0+H}$ via discriminators; compute $r^{\text{intr}}$ via RND
20:   World-model update: $\theta \leftarrow \theta - \rho\nabla(\mathcal{L}_h + \mathcal{L}_R + \mathcal{L}_Q)$
21:   Policy prior update: $\pi_\theta \leftarrow \pi_\theta - \rho\nabla\mathcal{L}_\pi$
22:   Discriminator update: $\delta_k \leftarrow \delta_k - \rho\nabla\mathcal{L}_{\text{disc}}$
23:   RND predictor update: $\hat{f} \leftarrow \hat{f} - \rho\nabla\mathcal{L}_{\text{RND}}$
24:   Update planning coefficient: $\alpha \leftarrow \text{schedule}(\mathcal{L}_{\text{disc}})$
25: **end for**

---

$f$, and $\sigma_{\text{intr}}$ is a running estimate of the standard deviation of intrinsic rewards. The predictor $\hat{f}$ is trained by minimizing the squared prediction error against the frozen target $f$:

$$\mathcal{L}_{\text{RND}} = \mathbb{E}_\mathcal{B}\left[\|\hat{f}(z') - f(z')\|_2^2\right] \tag{8}$$

## 4.2. Adaptive Uncertainty-Guided Planning

While intrinsic rewards enhance exploration in the policy prior and Q-function, adaptive uncertainty-guided mechanisms are needed to prevent excessive exploration. However, the world model cannot quantify its own prediction confidence, leading to overconfident planning that exploits erroneous predictions in OOD states. This problem becomes particularly severe in multi-stage tasks, where prediction errors accumulate across the planning horizon. To mitigate this problem, we employ an ensemble of dynamics models in latent space to quantify uncertainty and facilitate adaptive uncertainty-guided planning. The ensemble measures

uncertainty directly through the disagreement of its members, producing a single scalar that we use directly as a unified uncertainty estimate without distributional assumptions. This makes it more suitable for our online learning framework than offline methods. We apply ensembling exclusively to the dynamics model, as reward targets are non-stationary during joint training and reward ensembling would adversely affect policy learning.

We replace the world model dynamics network with an ensemble of $N_e$ independent networks $\bar{d}_\theta = \{d_{\theta_1}, d_{\theta_2}, \ldots, d_{\theta_{N_e}}\}$ that share the same architecture but are differently initialized. Given the current latent state $z_t$ and action $a_t$, each ensemble member produces a prediction $z'^{(n_e)}$, and the final prediction is obtained by averaging across all members as $\bar{z}_{t+1} = \frac{1}{N_e} \sum_{n_e=1}^{N_e} z'^{(n_e)}_{t+1}$. The disagreement among ensemble members serves as a proxy for uncertainty, where predictions converge in familiar state-action regions while diverging significantly in unfamiliar ones. We quantify this uncertainty using the pairwise disagreement metric:

$$\tilde{\mathcal{U}}_n(z_t, a_t) = \sqrt{\frac{2}{N_e(N_e-1)} \sum_{i=1}^{N_e} \sum_{j=i+1}^{N_e} \left\| z'^{(i)}_{t+1} - z'^{(j)}_{t+1} \right\|_2^2} \tag{9}$$

We incorporate this uncertainty estimate into the MPPI planning objective. The uncertainty-adjusted value is computed as:

$$\hat{Q}_{\text{adjusted}} = \hat{Q} + \alpha \cdot \sum_{h=0}^{H-1} \gamma^h \tilde{\mathcal{U}}_n(z_{t+h}, a_{t+h}) \tag{10}$$

where $\tilde{\mathcal{U}}_n$ represents the normalized uncertainty, and $\alpha$ controls the exploration-exploitation balance.

**Assumption 1 (Lipschitz Value Function).** The value function under the true latent MDP is $L_V$-Lipschitz with respect to the latent state (Luo et al., 2018; Asadi et al., 2018):

$$\left| V^\pi_{\mathcal{M}}(z_1) - V^\pi_{\mathcal{M}}(z_2) \right| \leq L_V \|z_1 - z_2\|_2, \quad \forall z_1, z_2 \in \mathcal{Z}. \tag{11}$$

**Assumption 2 (Uncertainty as Error Bound).** We denote $d^*$ as the true latent dynamics of the environment. The normalized ensemble disagreement $\tilde{\mathcal{U}}_n(z, a)$ upper bounds the one-step latent dynamics error:

$$\|\bar{d}_\theta(z, a) - d^*(z, a)\|_2 \leq c_1 \tilde{\mathcal{U}}_n(z, a) + \epsilon_d, \tag{12}$$

where $c_1 > 0$ is a problem-dependent constant and $\epsilon_d \geq 0$ is the irreducible model bias.

**Assumption 3 (Reward Convergence).** We denote $R^*$ as the true reward function. In the conservative planning phase, the learned reward predictor is uniformly accurate:

$$|R_\theta(z, a) - R^*(z, a)| \leq \epsilon_R, \quad \forall(z, a). \tag{13}$$

Let $d^\pi_{\bar{d}_\theta}$ denote the normalized discounted occupancy measure induced by policy $\pi$ under dynamics $\bar{d}_\theta$. We define the model error exposure of a policy as

$$\text{Exposure}(\pi) := \mathbb{E}_{(z,a) \sim d^\pi_{\bar{d}_\theta}} \left[ \|\bar{d}_\theta(z, a) - d^*(z, a)\|_2 \right], \tag{14}$$

which measures how much $\pi$ visits regions where the learned dynamics deviates from the true dynamics. We denote by $\hat{\mathcal{M}} = (\mathcal{Z}, \mathcal{A}, \bar{d}_\theta, R_\theta, \gamma)$ the learned model MDP induced by the ensemble-mean dynamics $\bar{d}_\theta$ and the learned reward predictor $R_\theta$.

**Theorem 1 (Conservative Error Transformation).** *For any policy $\pi$ with $\gamma \in [0, 1)$ and $L_V, \epsilon_R, \epsilon_d \geq 0$, under Assumptions 1 and 3, the two-sided bound satisfies:*

$$\left| J_{\mathcal{M}}(\pi) - J_{\hat{\mathcal{M}}}(\pi) \right| \leq \frac{\gamma L_V}{1-\gamma} \text{Exposure}(\pi) + \frac{\epsilon_R}{1-\gamma}. \tag{15}$$

*Under Assumptions 1–3, for $\alpha \leq -\gamma c_1 L_V$ with $c_1 > 0$, the one-sided bound satisfies:*

$$J_{\mathcal{M}}(\pi) \geq J_{\tilde{\mathcal{M}}}(\pi) - \epsilon_{\text{bias}}, \quad \epsilon_{\text{bias}} = \frac{\gamma L_V \epsilon_d + \epsilon_R}{1-\gamma}. \tag{16}$$

*Proof.* See Appendix A.

Given Theorem 1, it follows that vanilla model-based planning may overestimate or underestimate the true performance $J_{\mathcal{M}}(\pi)$, with the deviation scaling with policy exposure to model uncertainty. However, by introducing a sufficiently large uncertainty penalty $\alpha \leq -\gamma c_1 L_V$, the two-sided error is transformed into a one-sided conservative bias. This ensures that the conservative model $\tilde{\mathcal{M}}$ provides a lower bound on the true performance, i.e., $J_{\mathcal{M}}(\pi) \geq J_{\tilde{\mathcal{M}}}(\pi) - \epsilon_{\text{bias}}$. In practice, this conservative property discourages the agent from exploiting high-uncertainty regions where model predictions are unreliable, leading to more robust policy optimization.

**Theorem 2 (Exposure-Independent Bound).** *Under the conditions of Theorem 1, the uncertainty bound is strictly tighter than the vanilla bound whenever:*

$$\text{Exposure}(\pi) > \epsilon_d. \tag{17}$$

*Proof.* See Appendix A.

Given Theorem 2, it follows that when policy exposure exceeds the threshold $\epsilon_d$, the uncertainty-guided planning provides a strictly tighter performance guarantee. This advantage becomes more pronounced for exploratory policies or multi-stage tasks where cumulative model-error exposure is large.

Therefore, we adaptively adjust $\alpha$ across training stages. In early training, when the dense reward function is unreliable,

we set $\alpha > 0$ to bias planning toward less-visited regions for diverse exploration. As training progresses and the reward function stabilizes, we set $\alpha < 0$ so that uncertainty acts as a penalty, promoting conservative planning. We leverage the reward function loss $\mathcal{L}_{\text{disc}}$ as an indicator of both training progress and reward reliability. The coefficient $\alpha$ is adaptively scheduled as:

$$\alpha = \begin{cases} \alpha_{\text{bonus}} & \text{if } \mathcal{L}_{\text{disc}} > \tau_{\text{high}} \\ -\alpha_{\text{penalty}} & \text{if } \mathcal{L}_{\text{disc}} < \tau_{\text{low}} \\ \alpha_{\text{bonus}} - (\alpha_{\text{bonus}} + \alpha_{\text{penalty}}) \cdot \Omega & \text{otherwise} \end{cases} \tag{18}$$

where $\Omega = \frac{\tau_{\text{high}} - \mathcal{L}_{\text{disc}}}{\tau_{\text{high}} - \tau_{\text{low}}}$, and $\tau_{\text{high}}$ and $\tau_{\text{low}}$ are threshold values that define the transition region. This adaptive mechanism enables automatic progression from optimistic exploration to pessimistic planning as the reward function stabilizes.

### 4.3. Training Scheme

QUEST learns a dense reward function from limited demonstrations while jointly training all MBRL components. To further boost data efficiency, we introduce a bootstrap training scheme for meaningful uncertainty quantification across ensemble members and a hybrid sampling strategy to accelerate learning from rare successful transitions.

To obtain diverse ensemble members that produce informative disagreement signals, we employ bootstrap aggregation during training. Each ensemble dynamics member is trained on a randomly sampled subset of the training batch, where each transition is included with probability $p_b$. This bootstrap sampling introduces variation in the training data distribution seen by each member, causing them to learn slightly different dynamics functions. The ensemble dynamics loss is computed as:

$$\mathcal{L}_h = \frac{1}{N_e} \sum_{n_e=1}^{N_e} \mathbb{E}_{(s,a,s') \sim \mathcal{B}^{(n_e)}} \left\| d_{\theta_{n_e}}(z,a) - \text{sg}(h_\theta(s')) \right\|_2^2 \tag{19}$$

where $\mathcal{B}^{(n_e)}$ denotes the bootstrap subset for the $n_e$-th member and sg is the stop-grad operator.

In multi-stage manipulation tasks, successful stage transitions are rare events under sparse rewards and limited demonstrations. Uniform sampling from the replay buffer dilutes these valuable experiences among abundant failure cases, slowing down policy learning. To address this, we propose a hybrid sampling strategy that combines samples from multiple sources with stage-based prioritization, assigning higher sampling probability to transitions from later stages. During training, each batch is constructed using a fixed ratio of samples from two sources, including the prioritized replay buffer and the expert demonstration buffer. The full training procedure of QUEST is summarized in Algorithm 1.

## 5. Experiments

We evaluate our method across 16 challenging visual multi-stage manipulation tasks characterized by sparse rewards with limited demonstrations. This includes 5 manipulation tasks and 2 humanoid manipulation tasks with high-dimensional action spaces from ManiSkill3, 5 tasks from Meta-World, and 4 tasks from Robosuite. These tasks represent the most challenging scenarios from each domain. This reflects realistic settings where expert demonstrations are costly to obtain. Accordingly, we use only 10 expert demonstrations for all tasks. For fair comparison, we adopt the same environment settings, input configurations, and expert data collection procedures as Demo[3]. Additional experimental details are provided in Appendix E. Multi-stage task definitions and further implementation details are provided in Appendix F. Through these evaluations, we aim to answer the following questions:

1. Can our proposed method achieve stable policy learning with limited demonstrations in multi-stage tasks?

2. What is the relative importance of each algorithmic component, and how does performance scale with the amount of demonstration data?

3. Can the learned policy transfer to real-world robots?

### 5.1. Baselines

To assess the effectiveness of our proposed method, we compare it against four relevant approaches.

**Demo**[3] **(Escoriza et al., 2025)** is a model-based RL algorithm that leverages demonstrations to jointly learn a policy, world model, and online dense reward function. Built upon TD-MPC2, it achieves impressive results on sparse-reward tasks, particularly in ManiSkill.

**MoDem2 (Hansen et al., 2022a)** is an updated version of MoDem that replaces the TD-MPC backbone with TD-MPC2. It enhances sample efficiency through interactive learning with demonstration oversampling and serves as a strong baseline for demonstration-augmented world models but lacks online reward learning.

**LaNE (Zhao et al., 2025)** is a model-free RL method that jointly learns an online reward function using a pre-trained feature extractor to construct latent embeddings and incentivize exploration toward demonstration-similar regions. It represents the state-of-the-art for model-free approaches in sparse-reward settings.

**TD-MPC2 (Hansen et al., 2023)** is a robust model-based algorithm combining temporal difference learning with MPC. We include it to demonstrate pure MBRL performance without demonstration-augmented mechanisms for reward learning or uncertainty quantification.

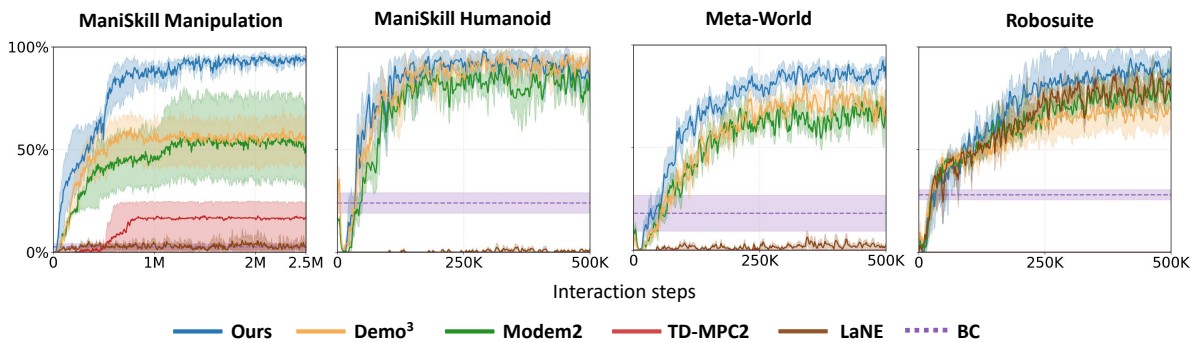

*Figure 3.* **Learning curves**. Success rate as a function of interaction steps for each of the four domains that we consider, averaged across all tasks and 5 random seeds. The shaded area indicates 95% stratified bootstrap confidence intervals based on IQM (Agarwal et al., 2021).

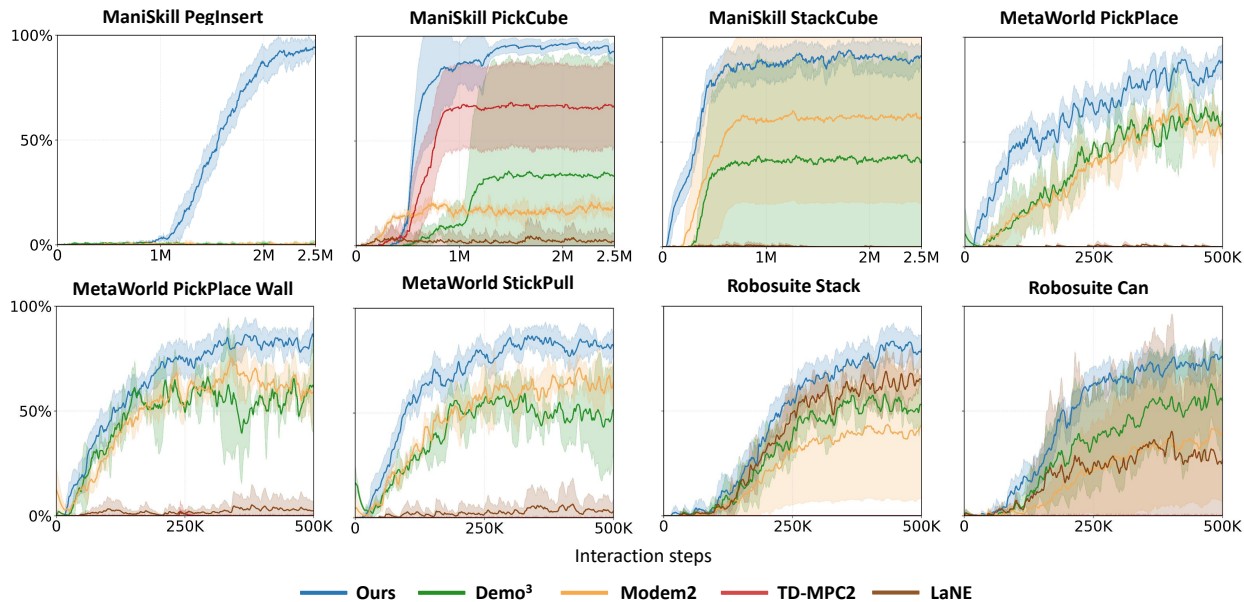

*Figure 4.* **Challenging tasks**. Success rate of our method and baselines on 8 hardest tasks selected across three domains (3 from ManiSkill, 3 from Meta-World, and 2 from Robosuite). Averaged across 5 random seeds. Shaded areas correspond to 95% confidence intervals.

## 5.2. Benchmark Results

As shown in Figure 3, we compare the data efficiency of our method against the baselines across all benchmarks. On average, QUEST achieves 17% higher success rate than the best-performing baseline. Notably, in ManiSkill3, our most challenging domain, QUEST achieves an average success rate of 97.2%, outperforming the second-best baseline by 38%. Furthermore, QUEST attains competitive performance on the high-dimensional action space tasks in ManiSkill Humanoids, while achieving the best performance on Meta-World and Robosuite tasks. Notably, QUEST not only outperforms on average across all tasks, but its advantage becomes more pronounced in the most challenging high-uncertainty tasks with only 10 expert demonstrations. Figure 4 shows learning curves for the most challenging task

of each benchmark. The PegInsert and Stack Cube tasks from ManiSkill require high precision and long planning horizons, resulting in high uncertainty given limited demonstrations. Additionally, PegInsert involves frequent contact interactions, where complex contact dynamics combined with domain randomization introduce significant aleatoric uncertainty. QUEST is the only algorithm that reliably solves both tasks within the interaction budget. Overall, QUEST demonstrates superior robustness and efficiency on difficult manipulation tasks with limited demonstrations.

## 5.3. Analysis

**Relative importance of each component.** Figure 5 shows the effect of removing intrinsic reward driven exploration, adaptive uncertainty-guided planning, and hybrid sampling

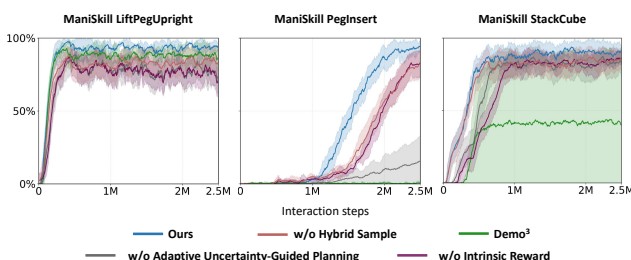

*Figure 5.* **Ablation analysis of individual method components across ManiSkill tasks**. Averaged across 5 random seeds. Shaded areas correspond to 95% confidence intervals.

strategy on the 3 manipulation tasks from ManiSkill3. Interestingly, adaptive uncertainty-guided planning brings a considerable improvement in performance, indicating the importance of balancing exploration and conservative planning throughout training. The effect of intrinsic reward driven exploration becomes more evident in multi-stage tasks, which require stronger exploration in the absence of dense reward signals. The hybrid sampling strategy improves sample efficiency by prioritizing rare successful transitions from later stages, preventing valuable experiences from being diluted by abundant failed demonstrations.

**Hyperparameter ablations.** As shown in Figure 6, we evaluate QUEST on the PegInsert task from ManiSkill with varying numbers of expert demonstrations. QUEST consistently requires fewer environment steps to reach 60% success rate, demonstrating superior sample efficiency compared to baseline methods. We further conduct ablation studies on the uncertainty coefficient $\alpha_{\text{penalty}}$ across two distinct task categories: robotic arm manipulation tasks and humanoid manipulation tasks. The results reveal that the uncertainty characteristics differ significantly between task types. These findings highlight the critical role of the uncertainty coefficient in balancing exploration and conservative planning, enabling adaptive behavior across diverse task domains.

**Wall-time comparison.** We compare wall-clock time across methods using the Robosuite environment. Table 1 reports hours per 500k interaction steps, averaged across 5 seeds. We also conduct an ablation study on the ensemble size $N_e$ to evaluate its effect on both performance and computational cost. QUEST is only slightly slower than Demo[3], and we attribute this overhead to the additional computation required for uncertainty quantification across multiple dynamics models. Using $N_e = 5$ ensemble members achieves a good balance between wall-clock time and success rate.

### 5.4. Real-world Experiments

**Real-world Setup.** We deploy five tasks from ManiSkill, including Pick Cube (Task A), Stack Cube (Task B), Place

*Table 1.* **Wall-time and Performance Comparison.** Hours per 500k interaction steps and Success Rate. Averaged across 5 seeds.

| Algorithm | Dynamics | Time (hours) ↓ | Success Rate ↑ |
|---|---|---|---|
| LaNE | No | 100.2 | $90.1 \pm 2.1$ |
| MoDem2 | No | 42.5 | $86.7 \pm 1.5$ |
| TD-MPC2 | No | **23.4** | $8.8 \pm 0.15$ |
| Demo[3] | No | 28.5 | $81.4 \pm 0.2$ |
| QUEST | 3 | 27.5 | $90.5 \pm 0.1$ |
| QUEST | 5 | 30.2 | $\mathbf{92.5 \pm 0.09}$ |
| QUEST | 8 | 38.8 | $91.2 \pm 0.09$ |

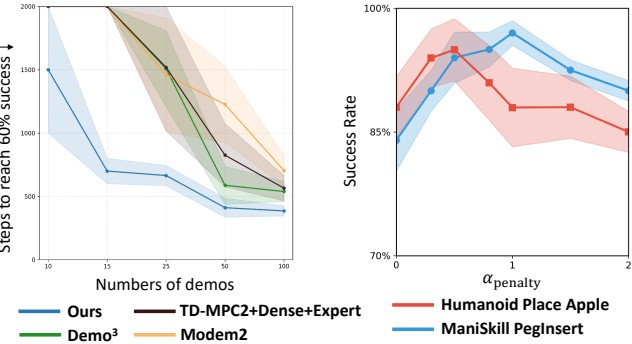

*Figure 6.* Left: Sample efficiency on ManiSkill PegInsert under varying numbers of expert demonstrations. Right: Ablation on the uncertainty coefficient $\alpha_{\text{penalty}}$ across two task categories.

Sphere (Task C), Pull Cube (Task D), and Poke Cube (Task E), to a real robot. We employ a simple but effective sim-to-real approach (see Figure 7). We perform hand-eye calibration on the cameras, configure the simulator with the corresponding camera positions, and capture scene images as background textures during training.

**Real-world Experimental Results.** We present the quantitative results in Table 2. QUEST achieves an average success rate of 62% across all five tasks, substantially outperforming BC (6%) and Demo[3] (32%). This validates the effectiveness of our sim-to-real approach in real-world manipulation. Additional experimental details are provided in the Appendix F.

*Table 2.* **Performance Comparison across Real-world Tasks.**

| METHOD | TASK A | TASK B | TASK C | TASK D | TASK E |
|---|---|---|---|---|---|
| BC | 2/10 | 0/10 | 1/10 | 0/10 | 0/10 |
| Demo[3] | 6/10 | 2/10 | 3/10 | 1/10 | 4/10 |
| **QUEST (Ours)** | **8/10** | **6/10** | **5/10** | **4/10** | **8/10** |

## 6. Related Work

**Learning from Demonstrations.** Learning from demonstrations helps mitigate the exploration challenge in sparse-reward environments and accelerates skill acquisition by leveraging expert guidance. One common approach is to use

demonstrations for imitation learning, which provides supervision signals for subsequent RL training (Kapturowski et al., 2018; Ball et al., 2023; Escontrela et al., 2022). Alternatively, during interactions, demonstrations can serve as on-policy regularization (Kang et al., 2018; Rajeswaran et al., 2017). Beyond these approaches, demonstrations can also be used to estimate reward functions for RL, enabling dense feedback from sparse environmental signals (Aytar et al., 2018; Vecerik et al., 2019; Xie et al., 2018; Singh et al., 2019; Zolna et al., 2020). However, these methods typically require extensive demonstrations to learn an accurate reward function, otherwise struggling with poor predictions on unseen states. We address this by jointly training the reward function and policy within an MBRL framework.

**Uncertainty Quantification and Exploration.** Standard deep reinforcement learning algorithms lack uncertainty quantification in both the agent and environment, leading to brittleness and poor performance in novel situations. Building uncertainty-guided agents is therefore important for developing robust and versatile systems (Yu et al., 2020b; Mavor-Parker et al., 2022). Parametric uncertainty estimation methods, such as bootstrap ensembles (An et al., 2021; Kidambi et al., 2020; Sun et al., 2023; Chua et al., 2018), Monte Carlo Dropout (Gal & Ghahramani, 2016), and randomized priors (Osband et al., 2018), may be susceptible to poor model specification and are most effective when dealing with large datasets (Kim & Oh, 2023; Tennenholtz & Mannor, 2022). In contrast, nonparametric methods such as k-nearest neighbors (Zhang et al., 2023; Fathabadi et al., 2022; Medina, 2013) are beneficial in regions of limited data, but require a proper distance metric. The quantified uncertainty is commonly leveraged for exploration, via curiosity-driven intrinsic motivation (Pathak et al., 2017; Suh et al., 2023; Sun et al., 2021). In this work, we employ a latent-space ensemble of dynamics models to quantify uncertainty and adaptively balance exploration and conservative planning throughout training. Compared with Bayesian or MC dropout methods that depend on distributional assumptions and large datasets, our ensemble measures uncertainty directly through member disagreement and remains compatible with the online learning setting under limited demonstrations. Unlike curiosity-driven approaches that add intrinsic rewards to the total reward and distort world-model learning, we restrict the intrinsic signal to the Q-function update and schedule the exploration-exploitation balance with a discriminator loss that tracks learning progress. These design choices let a single uncertainty signal drive both exploration and conservative planning in QUEST.

## 7. Conclusion

We present QUEST to address robotic manipulation with limited demonstrations by adaptively balancing exploration

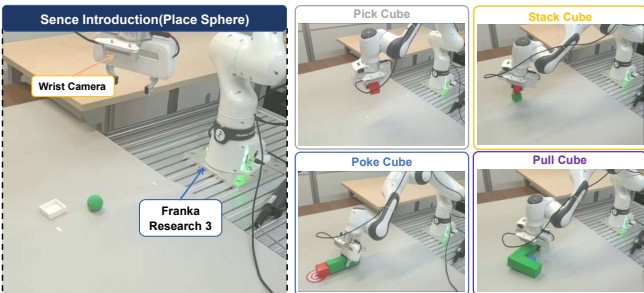

*Figure 7.* **Real-world robot platform**.

and exploitation via uncertainty quantification. Leveraging RND for intrinsic rewards, ensemble dynamics for planning, and hybrid sampling, QUEST outperforms state-of-the-art methods across multiple benchmarks. Real-world validation further confirms its practical applicability through successful zero-shot sim-to-real transfer.

While QUEST shows strong performance, several limitations remain. Our real-world evaluation relies on sim-to-real transfer, limiting task diversity to accurately simulatable scenarios. Future work could explore training with real-world teleoperated demonstrations mapped to simulation. Additionally, QUEST still depends on sparse environment rewards; leveraging large language or vision-language models for reward signals could enable fully reward-free learning.

## Impact Statement

This paper presents work whose goal is to advance the field of robot learning. While there are many potential societal consequences in developing embodied intelligence for the real world, we do not feel our work presents any particular implications that should be highlighted here.

## Acknowledgements

The work was supported in part by Fundamental and Interdisciplinary Disciplines Breakthrough Plan of the Ministry of Education of China under grant No. JYB2025XDXM210, NSFC under grant No. 62125305, No. U23A20339, No. 62573339, No. 62503380, and Natural Science Foundation of Shaanxi Province under Grant 2025SYSSYSZD-083, State Grid Corporation of China Headquarters Science and Technology Project under Grant No. 52060025005B-439-ZN, Shaanxi Provincial Science Fund for Distinguished Young Scholars under Grant No. 2025JC-JCQN-077, and CIE–Tencent Robotics X Rhino-Bird Focused Research Program.

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

# A. Proofs

## A.1. Preliminaries and Notation

We consider a latent MDP with latent state space $\mathcal{Z}$ and action space $\mathcal{A}$. Let the true latent MDP be

$$\mathcal{M} = (\mathcal{Z}, \mathcal{A}, d^*, R^*, \gamma), \qquad \gamma \in [0, 1). \tag{20}$$

We denote by $\bar{d}_\theta$ the ensemble-mean latent dynamics model and by $R_\theta$ the learned reward predictor. The learned model MDP is

$$\hat{\mathcal{M}} = (\mathcal{Z}, \mathcal{A}, \bar{d}_\theta, R_\theta, \gamma). \tag{21}$$

In the conservative planning phase, we define the uncertainty-penalized reward as

$$\tilde{R}(z, a) = R_\theta(z, a) + \alpha \tilde{\mathcal{U}}_n(z, a), \tag{22}$$

and the corresponding conservative model MDP

$$\tilde{\mathcal{M}} = (\mathcal{Z}, \mathcal{A}, \bar{d}_\theta, \tilde{R}, \gamma). \tag{23}$$

For any policy $\pi$, let $J_{\mathcal{X}}(\pi)$ denote the expected discounted return under MDP $\mathcal{X} \in \{\mathcal{M}, \hat{\mathcal{M}}, \tilde{\mathcal{M}}\}$:

$$J_{\mathcal{X}}(\pi) = \mathbb{E}\Big[ \sum_{t=0}^{\infty} \gamma^t R_{\mathcal{X}}(z_t, a_t) \Big], \quad a_t \sim \pi(\cdot|z_t), \tag{24}$$

where transitions follow the dynamics in $\mathcal{X}$.

Let $d_d^\pi(z, a)$ denote the normalized discounted occupancy measure induced by $\pi$ under dynamics $d$:

$$d_d^\pi(z, a) \triangleq (1 - \gamma) \sum_{t=0}^{\infty} \gamma^t \Pr(z_t = z, a_t = a \mid \pi, d). \tag{25}$$

Then for any measurable function $g : \mathcal{Z} \times \mathcal{A} \to \mathbb{R}$, we have the standard identity:

$$\mathbb{E}_{\pi, d}\Big[ \sum_{t=0}^{\infty} \gamma^t g(z_t, a_t) \Big] = \frac{1}{1 - \gamma} \mathbb{E}_{(z,a) \sim d_d^\pi}\big[ g(z, a) \big]. \tag{26}$$

We define the model error exposure of a policy $\pi$ as:

$$\text{Exposure}(\pi) := \mathbb{E}_{(z,a) \sim d_{\bar{d}_\theta}^\pi} \big[ \| \bar{d}_\theta(z, a) - d^*(z, a) \|_2 \big]. \tag{27}$$

## A.2. Proof of Theorem 1

**Proof.** Fix an arbitrary policy $\pi$. Define the one-step *Bellman residual* evaluated on the true value function:

$$\Delta(z) := \mathbb{E}_{a \sim \pi(\cdot|z)}\Big[ R^*(z, a) + \gamma V_{\mathcal{M}}^\pi(d^*(z, a)) - \big( R_\theta(z, a) + \gamma V_{\mathcal{M}}^\pi(\bar{d}_\theta(z, a)) \big) \Big]. \tag{28}$$

By triangle inequality and Assumption 1,

$$|\Delta(z)| \leq \mathbb{E}_{a \sim \pi(\cdot|z)}\Big( |R^*(z, a) - R_\theta(z, a)| + \gamma |V_{\mathcal{M}}^\pi(d^*(z, a)) - V_{\mathcal{M}}^\pi(\bar{d}_\theta(z, a))| \Big)$$

$$\leq \mathbb{E}_{a \sim \pi(\cdot|z)}\Big( |R^*(z, a) - R_\theta(z, a)| + \gamma L_V \| d^*(z, a) - \bar{d}_\theta(z, a) \|_2 \Big). \tag{29}$$

Next, consider rolling out $\pi$ under the model dynamics $\bar{d}_\theta$ starting from the same initial distribution. A standard telescoping argument for policy evaluation under mismatched models (via repeated substitution of the Bellman equation) yields that the return difference can be written as the discounted sum of residuals:

$$J_{\mathcal{M}}(\pi) - J_{\hat{\mathcal{M}}}(\pi) = \mathbb{E}_{\pi, \bar{d}_\theta}\Big[ \sum_{t=0}^{\infty} \gamma^t \Delta(z_t) \Big]. \tag{30}$$

Taking absolute values and applying (29) together with (26) gives:

$$
\begin{aligned}
\left|J_{\mathcal{M}}(\pi) - J_{\hat{\mathcal{M}}}(\pi)\right| &\leq \mathbb{E}_{\pi,\bar{d}_\theta}\Big[\sum_{t=0}^\infty \gamma^t |\Delta(z_t)|\Big] \\
&\leq \mathbb{E}_{\pi,\bar{d}_\theta}\Big[\sum_{t=0}^\infty \gamma^t \Big(|R^*(z_t,a_t) - R_\theta(z_t,a_t)| + \gamma L_V \|d^*(z_t,a_t) - \bar{d}_\theta(z_t,a_t)\|_2\Big)\Big] \\
&= \frac{1}{1-\gamma}\,\mathbb{E}_{(z,a)\sim d_{\bar{d}_\theta}^\pi}\Big[|R^*(z,a) - R_\theta(z,a)|\Big] + \frac{\gamma L_V}{1-\gamma}\,\mathbb{E}_{(z,a)\sim d_{\bar{d}_\theta}^\pi}\Big[\|d^*(z,a) - \bar{d}_\theta(z,a)\|_2\Big] \\
&= \frac{1}{1-\gamma}\,\mathbb{E}_{d_{\bar{d}_\theta}^\pi}\Big[|R^* - R_\theta|\Big] + \frac{\gamma L_V}{1-\gamma}\,\mathrm{Exposure}(\pi).
\end{aligned}
\tag{31}
$$

Finally, by Assumption 3, $\mathbb{E}_{d_{\bar{d}_\theta}^\pi}[|R^* - R_\theta|] \leq \epsilon_R$. Substituting into (31) yields (15).

Then, starting from the two-sided bound (15), we obtain a lower bound:

$$
J_{\mathcal{M}}(\pi) \geq J_{\hat{\mathcal{M}}}(\pi) - \frac{\gamma L_V}{1-\gamma}\,\mathrm{Exposure}(\pi) - \frac{\epsilon_R}{1-\gamma}.
\tag{32}
$$

Using Assumption 2 and the definition of $\mathrm{Exposure}(\pi)$,

$$
\begin{aligned}
\mathrm{Exposure}(\pi) &= \mathbb{E}_{(z,a)\sim d_{\bar{d}_\theta}^\pi}\Big[\|\bar{d}_\theta(z,a) - d^*(z,a)\|_2\Big] \\
&\leq \mathbb{E}_{d_{\bar{d}_\theta}^\pi}\Big[c_1 \tilde{\mathcal{U}}_n(z,a) + \epsilon_d\Big] = c_1\,\mathbb{E}_{d_{\bar{d}_\theta}^\pi}[\tilde{\mathcal{U}}_n(z,a)] + \epsilon_d.
\end{aligned}
\tag{33}
$$

Substituting (33) into (32) yields:

$$
J_{\mathcal{M}}(\pi) \geq J_{\hat{\mathcal{M}}}(\pi) - \frac{\gamma L_V c_1}{1-\gamma}\,\mathbb{E}_{d_{\bar{d}_\theta}^\pi}[\tilde{\mathcal{U}}_n] - \frac{\gamma L_V \epsilon_d + \epsilon_R}{1-\gamma}.
\tag{34}
$$

Next, note that $\hat{\mathcal{M}}$ and $\tilde{\mathcal{M}}$ share the same dynamics $\bar{d}_\theta$. By definition of $\tilde{R}$ in (22) and the occupancy identity (26),

$$
\begin{aligned}
J_{\tilde{\mathcal{M}}}(\pi) &= \mathbb{E}_{\pi,\bar{d}_\theta}\Big[\sum_{t=0}^\infty \gamma^t\big(R_\theta(z_t,a_t) + \alpha\tilde{\mathcal{U}}_n(z_t,a_t)\big)\Big] \\
&= J_{\hat{\mathcal{M}}}(\pi) + \alpha\,\mathbb{E}_{\pi,\bar{d}_\theta}\Big[\sum_{t=0}^\infty \gamma^t \tilde{\mathcal{U}}_n(z_t,a_t)\Big] \\
&= J_{\hat{\mathcal{M}}}(\pi) + \frac{\alpha}{1-\gamma}\,\mathbb{E}_{d_{\bar{d}_\theta}^\pi}[\tilde{\mathcal{U}}_n(z,a)].
\end{aligned}
\tag{35}
$$

Rearranging (35) gives:

$$
J_{\hat{\mathcal{M}}}(\pi) = J_{\tilde{\mathcal{M}}}(\pi) - \frac{\alpha}{1-\gamma}\,\mathbb{E}_{d_{\bar{d}_\theta}^\pi}[\tilde{\mathcal{U}}_n].
\tag{36}
$$

Substituting (36) into (34) yields:

$$
\begin{aligned}
J_{\mathcal{M}}(\pi) &\geq J_{\tilde{\mathcal{M}}}(\pi) - \frac{\alpha}{1-\gamma}\mathbb{E}_{d_{\bar{d}_\theta}^\pi}[\tilde{\mathcal{U}}_n] - \frac{\gamma L_V c_1}{1-\gamma}\mathbb{E}_{d_{\bar{d}_\theta}^\pi}[\tilde{\mathcal{U}}_n] - \frac{\gamma L_V \epsilon_d + \epsilon_R}{1-\gamma} \\
&= J_{\tilde{\mathcal{M}}}(\pi) - \frac{\alpha + \gamma c_1 L_V}{1-\gamma}\mathbb{E}_{d_{\bar{d}_\theta}^\pi}[\tilde{\mathcal{U}}_n] - \frac{\gamma L_V \epsilon_d + \epsilon_R}{1-\gamma}.
\end{aligned}
\tag{37}
$$

If $\alpha \leq -\gamma c_1 L_V$, then the middle term in (37) is non-negative and can be dropped, yielding:

$$
J_{\mathcal{M}}(\pi) \geq J_{\tilde{\mathcal{M}}}(\pi) - \frac{\gamma L_V \epsilon_d + \epsilon_R}{1-\gamma}.
\tag{38}
$$

This proves (16) with $\epsilon_{\mathrm{bias}} = \frac{\gamma L_V \epsilon_d + \epsilon_R}{1-\gamma}$.

### A.3. Proof of Theorem 2

**Proof.**   From Theorem 1, the *vanilla* guarantee induced by (15) can be written as:

$$J_{\mathcal{M}}(\pi) \geq J_{\hat{\mathcal{M}}}(\pi) - \Delta_{\text{van}}(\pi), \qquad \Delta_{\text{van}}(\pi) = \frac{\gamma L_V}{1 - \gamma} \text{Exposure}(\pi) + \frac{\epsilon_R}{1 - \gamma}. \tag{39}$$

The *uncertainty-guided conservative* guarantee in (16) can be written as:

$$J_{\mathcal{M}}(\pi) \geq J_{\tilde{\mathcal{M}}}(\pi) - \Delta_{\text{unc}}, \qquad \Delta_{\text{unc}} = \epsilon_{\text{bias}} = \frac{\gamma L_V \epsilon_d + \epsilon_R}{1 - \gamma}. \tag{40}$$

We say the uncertainty-guided bound is *strictly tighter in terms of the gap* if $\Delta_{\text{unc}} < \Delta_{\text{van}}(\pi)$. Comparing (39) and (40),

$$\begin{aligned} \Delta_{\text{unc}} < \Delta_{\text{van}}(\pi) &\iff \frac{\gamma L_V \epsilon_d + \epsilon_R}{1 - \gamma} < \frac{\gamma L_V}{1 - \gamma} \text{Exposure}(\pi) + \frac{\epsilon_R}{1 - \gamma} \\ &\iff \gamma L_V \epsilon_d + \epsilon_R < \gamma L_V \text{Exposure}(\pi) + \epsilon_R \qquad (\text{since } 1 - \gamma > 0) \\ &\iff \gamma L_V \epsilon_d < \gamma L_V \text{Exposure}(\pi) \\ &\iff \epsilon_d < \text{Exposure}(\pi) \qquad (\gamma L_V > 0). \end{aligned}$$

This proves that the gap is strictly smaller whenever (17) holds.

# B. Additional Results

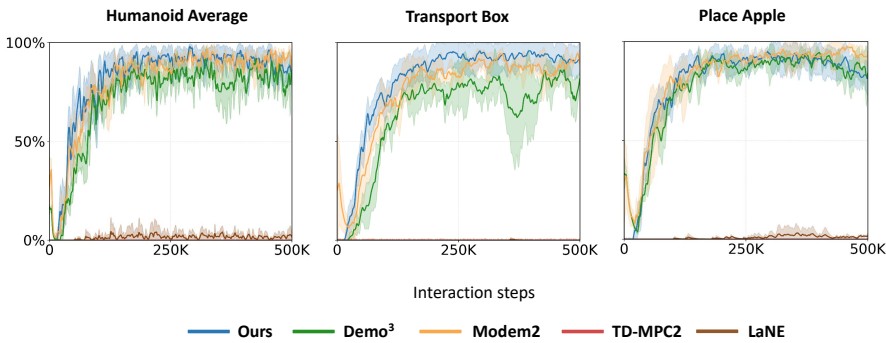

*Figure 8.* Experimental results for the Humanoid benchmark in ManiSkill, using five random seeds.

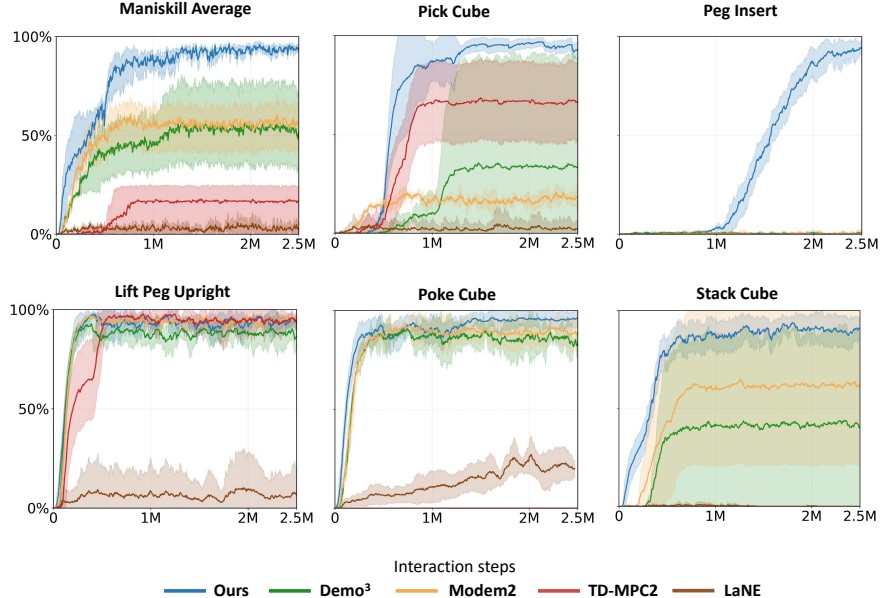

*Figure 9.* Experimental results for the Manipulation benchmark in ManiSkill, using five random seeds.

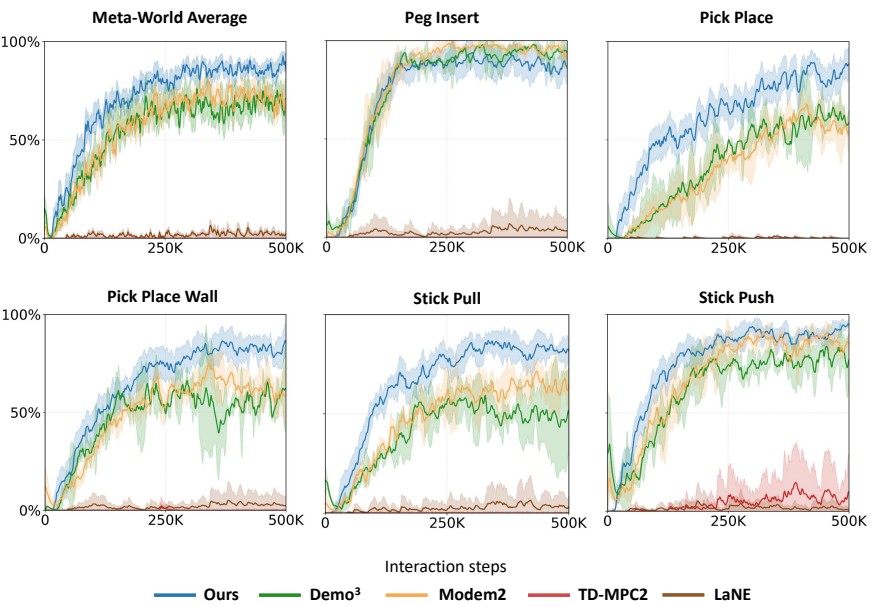

*Figure 10.* Experimental results for the Meta-World benchmark, using five random seeds.

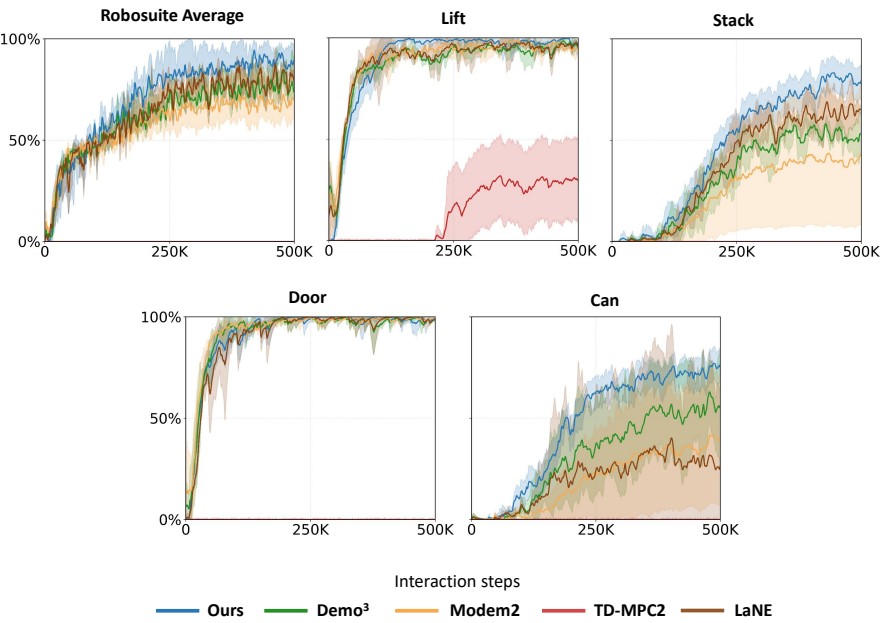

*Figure 11.* Experimental results for the Robosuite benchmark, using five random seeds.

# C. Baselines

## C.1. Baseline Implementations

In this section, we provide a detailed overview of the baseline algorithms used for comparison in our experiments.

**Demo**[3] (**Escoriza et al., 2025**) is a model-based RL algorithm that utilizes demonstrations to simultaneously learn a policy, a world model, and an online dense reward function. It builds upon TD-MPC2, coupling exploration signals with reward learning and using MPPI for action planning. Demo[3] achieves impressive results on sparse-reward tasks, particularly in the ManiSkill environment. We utilized the official implementation of Demo[3][1], with all hyperparameter settings aligned with it.

**MoDem2 (Hansen et al., 2022a)** is an updated version of MoDem that replaces the TD-MPC backbone with TD-MPC2. It improves sample efficiency through interactive learning with oversampling of demonstration data. MoDem2 serves as a strong baseline for demonstration-augmented world models but lacks online reward learning. We adapted the official MoDem2 implementation[2] to our benchmark, keeping its default hyperparameter settings throughout. To accommodate multi-image inputs, we added an additional encoder and averaged its output embeddings.

**LaNE (Zhao et al., 2025)** is a model-free RL method that jointly learns an online reward function. It employs a pre-trained feature extractor to construct a latent embedding space and incentivizes the agent to explore regions similar to the demonstrations under similarity-based rewards. LaNE represents the state-of-the-art for model-free approaches in sparse-reward settings such as Robosuite.

**TD-MPC2 (Hansen et al., 2023)** is a robust model-based algorithm combining temporal difference learning with Model Predictive Control (MPC). We include it to demonstrate the performance of a pure MBRL algorithm without the specific demonstration-augmented mechanisms for reward learning or uncertainty quantification. We employed the official implementation of TD-MPC2[3], with all hyperparameter settings aligned with its configuration.

---

[1]https://github.com/adrialopezescoriza/demo3
[2]https://github.com/facebookresearch/modemv2
[3]https://github.com/nicklashansen/tdmpc2

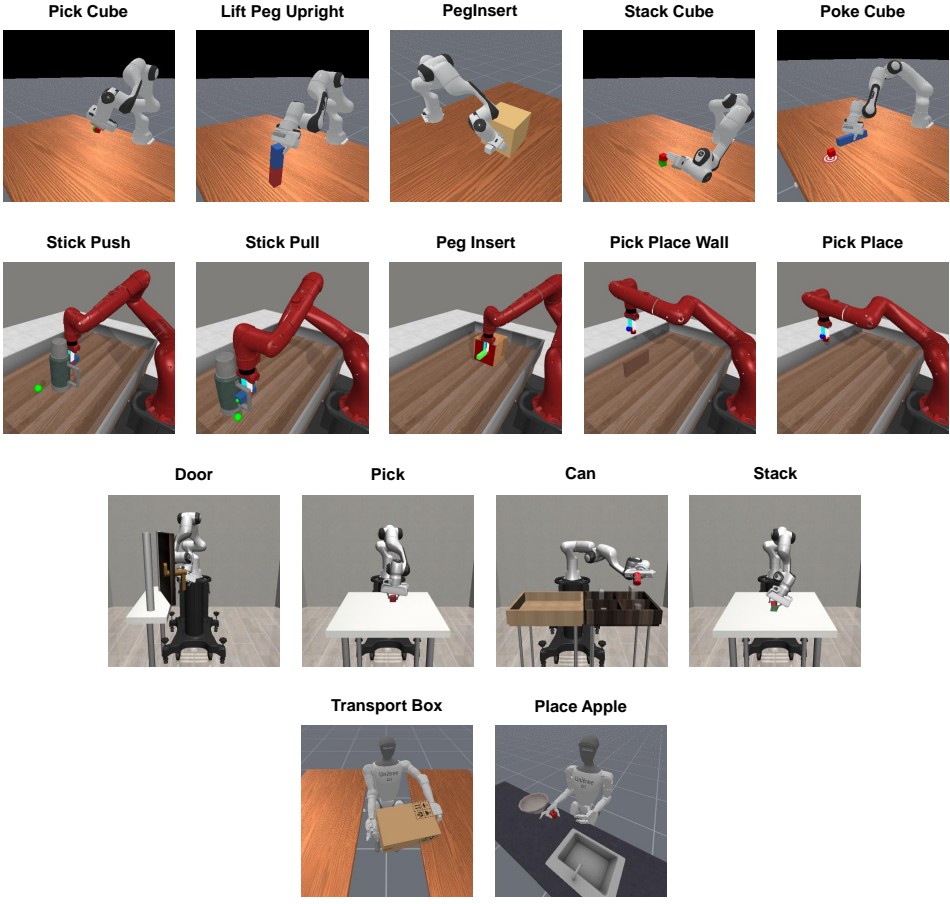

*Figure 12.* Visual representation of all tasks, listed top-to-bottom: ManiSkill, Meta-World, Robosuite, ManiSkill Humanoid

## D. Expert Demonstrations

Our approach and the baseline requiring expert data both obtain expert data by training a TD-MPC2 model. During training, the TD-MPC2 model takes RGB observations and proprioceptive states from the environment as input. The Meta-World environment provides a single viewpoint, while the Robosuite environment offers only RGB observations without states. The trained model is tested in a stage-based environment, where image observations, proprioceptive states, and sparse stage rewards from successful sample processes are recorded as expert data. For all model training, we utilize 10 expert data points.

# E. Experiment Details

## E.1. Real-world Implementation

We deploy five tasks from ManiSkill, including Pick Cube (Task A), Stack Cube (Task B), Place Sphere (Task C), Pull Cube (Task D), and Poke Cube (Task E), to a real Franka Research 3 robot. To achieve zero-shot sim-to-real transfer without domain randomization or real-world fine-tuning, we employ a simple but effective visual alignment approach. The key insight is that the domain gap between simulation and reality primarily arises from two factors: camera perspective misalignment and background appearance discrepancy. To address the first factor, we perform hand-eye calibration, then configure the virtual camera in the simulator with identical intrinsic and extrinsic parameters, ensuring that objects at the same physical location appear at consistent pixel coordinates across both domains. For the second factor, we capture reference images of the real workspace background before training and integrate them as environment textures in the simulator. During training, the rendered robot and objects are overlaid onto these background images through depth-based compositing. This approach allows the agent to observe backgrounds that are visually identical to the deployment environment, effectively eliminating background-induced domain shift while implicitly encoding real-world lighting conditions and sensor characteristics. With these two alignment strategies, the policy trained entirely in simulation can be directly deployed to the real robot without any adaptation, achieving effective zero-shot transfer.

## E.2. Challenging Tasks

Among all experimental tasks, certain tasks exhibit significantly higher difficulty due to the inherent uncertainty in both manipulation dynamics and environmental configurations. This uncertainty arises from two primary sources: (1) the requirement for models to perform complex, contact-rich manipulations where subtle variations in force and position can lead to drastically different outcomes, and (2) highly randomized initial states that demand robust generalization across diverse scenarios. For example, the PegInsert task in the ManiSkill benchmark exemplifies such high-uncertainty challenges compared to its Meta-World counterpart. As illustrated in Figure 13, ManiSkill introduces substantial stochasticity in multiple dimensions. The position and orientation of pegs vary significantly across episodes, while the size and location of target holes are also randomized. This compounded uncertainty makes accurate prediction inherently difficult, as the world model must capture a broader distribution of possible state transitions rather than a narrow, deterministic mapping, thereby placing greater demands on the model's ability to quantify and leverage uncertainty during planning.

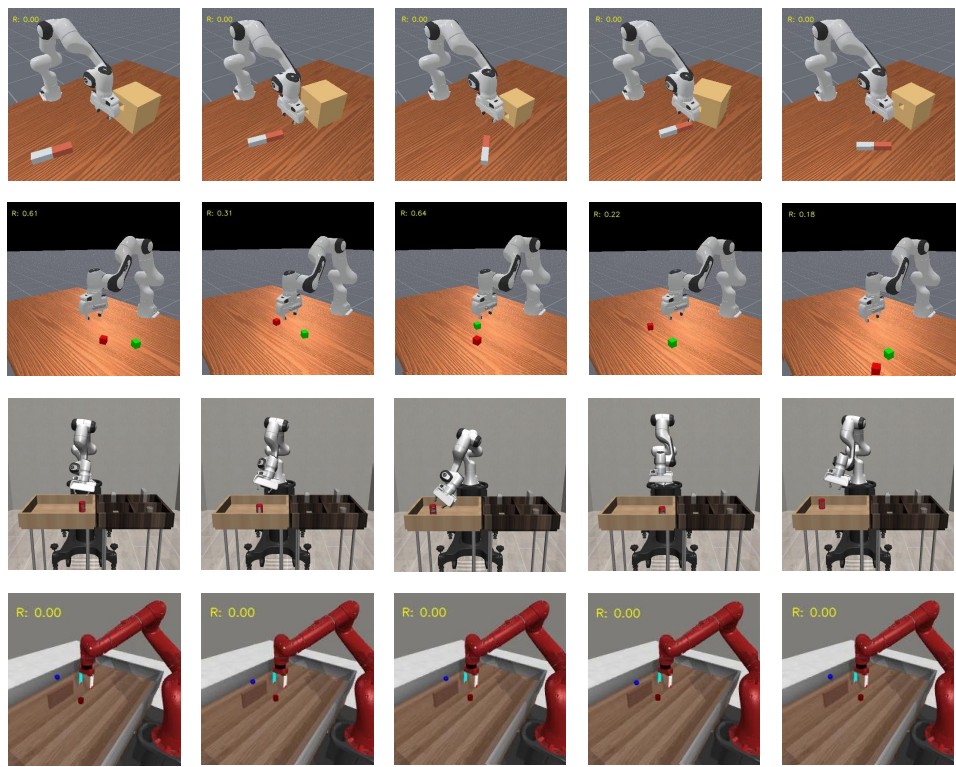

*Figure 13.* Task Difficulty Comparison via Randomization Ranges. Each row shows five randomized initial configurations of a single task: ManiSkill PegInsert (Row 1) and StackCube (Row 2), and their counterparts in Robosuite (Row 3) and Meta-World (Row 4). The ManiSkill tasks span substantially wider ranges in object position, orientation, and size than the Robosuite and Meta-World counterparts, resulting in higher task difficulty under limited demonstrations.

# F. Implementation Details

## F.1. Model Architecture

*Table 3.* Ensemble World Model Architecture

| Component | Architecture | Details |
|---|---|---|
| **Encoder** | ShiftAug $\rightarrow$ PixelPreprocess
Conv2d(3, 32, 7×7, s=2) $\rightarrow$ ReLU
Conv2d(32, 32, 5×5, s=2) $\rightarrow$ ReLU
Conv2d(32, 32, 3×3, s=2) $\rightarrow$ ReLU
Conv2d(32, 32, 3×3, s=2) $\rightarrow$ ReLU
Conv2d(32, 32, 3×3, s=1)
Flatten $\rightarrow$ Linear(512, 512)
SimNorm(dim=8) | RGB input |
| **Dynamics** | NormedLinear(519, 512, Mish)
NormedLinear(512, 512, Mish)
NormedLinear(512, 512, SimNorm) | Ensemble of 5 |
| **Reward** | NormedLinear(519, 512, Mish)
NormedLinear(512, 512, Mish)
Linear(512, 101) | Distributional |
| **Policy Prior** | NormedLinear(512, 512, Mish)
NormedLinear(512, 512, Mish)
Linear(512, 14) | 7D action $\times$ 2 (mean, std) |
| **Q-functions** | NormedLinear(519, 512, Mish, drop=0.01)
NormedLinear(512, 512, Mish)
Linear(512, 101) | Vectorized, 5 heads |
| **Discriminator** | Linear(512, 32) $\rightarrow$ Sigmoid
Linear(32, 1) | 3 networks |

## F.2. Definition of $K$-Stage in Multi-Stage Tasks

For the multi-stage manipulation tasks considered in this work, task progress is represented by task-specific semantic stages. Table 4 summarizes the stage definitions used for stage labeling and discriminator training. A dash indicates that no additional intermediate stage is used.

## F.3. Hyperparameters

The parameters used in our model are largely consistent with those of our baseline algorithm Demo[3]. Table 5 presents some key parameters of our algorithm.

*Table 4.* Definitions of intermediate stages and success criteria for multi-stage manipulation tasks.

| Task | Stage 1 | Stage 2 | Success Criteria |
|------|---------|---------|------------------|
| ***ManiSkill Manipulation*** | | | |
| Pick Cube | Object grasped | Near target | Object at goal |
| Lift Peg Upright | Peg grasped | Nearly vertical | Peg upright on table |
| PegInsert | Peg grasped | Aligned with socket | Peg inserted |
| Stack Cube | Top cube grasped | Above base cube | Cubes stacked |
| Poke Cube | Tool grasped | Cube contacted | Cube at goal |
| ***Meta-World*** | | | |
| Stick Push | Stick grasped | – | Object pushed to goal |
| Stick Pull | Stick grasped | – | Object pulled to goal |
| Peg Insert | Peg grasped | – | Peg inserted |
| Pick Place Wall | Object grasped | Above wall | Object beyond wall at goal |
| Pick Place | Object grasped | – | Object at goal |
| ***ManiSkill Humanoids*** | | | |
| Transport Box | Box grasped | Above target table | Box on target table |
| Place Apple | Apple grasped | Above bowl | Apple in bowl |
| ***Robosuite*** | | | |
| Door | Handle reached | – | Door opened |
| Pick | Object lifted | – | Object picked up |
| Can | Can grasped | – | Can at target |
| Stack | Top cube grasped | – | Cubes stacked |

*Table 5.* Hyperparameters used in the training setup.

| Hyperparameter | Value |
|---|---|
| **Replay buffer** | |
| Capacity | 300,000 |
| Sampling | Hybrid |
| **Architecture (5M)** | |
| Encoder arch. | ConvNet (image inputs) MLP (state inputs) |
| Conv. layers | 5 |
| Encoder MLP dim | 256 |
| Dynamics MLP dim | 512 |
| Latent state dim | 512 |
| Task embedding dim | 96 |
| $N_e$ | 5 |
| **Optimization** | |
| Update-to-data ratio | 1 |
| Batch size | 256 |
| Joint-embedding coef. | 20 |
| Reward prediction coef. | 0.1 |
| Value prediction coef. | 0.1 |
| Temporal coef. ($\lambda$) | 0.5 |
| $Q$-fn. momentum coef. | 0.99 |
| Policy prior entropy coef. | $1 \times 10^{-4}$ |
| Optimizer | Adam |
| Learning rate | $3 \times 10^{-4}$ |
| Encoder learning rate | $1 \times 10^{-4}$ |
| **Reward learning** | |
| Discriminator architecture | MLP |
| MLP dim | 32 |
| Discriminator learning rate | $3 \times 10^{-4}$ |
| Discriminator optimizer | Adam |
| Batch size | 256 |
| $\beta$ | 1/3 |
| Demo. sampling ratio | 50% |
| $\alpha_{\text{bonus}}$ | 0.3 |
| $\alpha_{\text{penalty}}$ | 0.8 |

