# OpenReview forum: "Uncertainty-Guided Exploration and Stable Planning for Sparse-Reward Manipulation from Limited Demonstrations"
_ICML.cc/2026/Conference — ICML 2026 regular_

### Official Review · Reviewer_T6hZ · 2026-03-09

**Soundness:** 3
**Presentation:** 3
**Significance:** 2
**Originality:** 2
**Overall Recommendation:** 4
**Confidence:** 4

**Summary:**

The proposed method algorithm QUEST is a model-based reinforcement learning framework that improves robotic manipulation with sparse rewards and limited demonstrations. QUEST adaptively switches between exploration and exploitation based on quantified uncertainty to achieve stable and efficient learning. It employs random network distillation to generate intrinsic rewards for the critic, enabling better modeling of environmental stochasticity, and uses ensemble dynamics to capture epistemic uncertainty for uncertainty-guided planning. Additionally, a hybrid sampling strategy prioritizes rare but successful stage transitions to accelerate multi-stage learning. The method demonstrates strong performance across various simulated manipulation benchmarks (MetaWorld, RoboSuite, and ManiSkill3), where it outperforms state-of-the-art baselines by up to 60% on complex tasks, and further achieves successful sim-to-real transfer on three real-world robotic tasks.

**Compliance With Llm Reviewing Policy:**

Affirmed.

**Key Questions For Authors:**

1. What is the novel contribution of the paper regarding exploration via intrinsic rewards, given that curiosity-driven reinforcement learning is already well-studied and not sufficiently reviewed in the paper?

2. How is transition uncertainty properly defined and justified, since replacing aleatoric uncertainty with model-environment mismatch may confuse true uncertainty with modeling errors?

3. Can the authors clarify the proposed algorithm, especially the missing or incomplete loss definitions that make the method section not self-contained?

4. Why are the real-world experiments limited and weakly justified, covering only very basic tasks, relying on nonstandard hand-eye calibration, and comparing only with behavior cloning?

**Limitations:**

yes.

**Strengths And Weaknesses:**

### Strength:

1. The structure of the paper is clear and easy to follow. The major contribution is well-presented in a reader-friendly format.
2. This paper studies an important problem. The development of the exploration algorithm is a key problem in RL. In the meantime, studying uncertainty in RL has been popular in RL in the recent years.
3. The proposed method is well presented in a technically sound manner.
4. The theoretical results provide important insight into the optimality of the proposed model-based planning algorithm.


### Weakness
1. My main concern lies in the novelty of the proposed intuition regarding exploration via intrinsic reward. To the best of my knowledge, curiosity-driven reinforcement learning, in which curiosity serves as a form of intrinsic reward, has already been extensively studied [1]. While the paper proposes an application of curiosity mechanisms, it lacks a sufficient literature review of existing methods, making it difficult to justify the claimed novelty.
Furthermore, using an ensemble of models (for example, ensemble Q-networks) to capture uncertainty is a well-established approach, initially proposed several years ago and followed by numerous studies.

[1] Pathak, Deepak, et al. “Curiosity-driven exploration by self-supervised prediction.” International Conference on Machine Learning. PMLR, 2017. (Cited over 3,000 times, this work has many following studies)

2. The paper’s definition of transition uncertainty, as the mismatch between the learned deterministic dynamics model and the stochastic environment transition, appears problematic and requires further justification. This definition implicitly replaces aleatoric uncertainty, without which the measurement of uncertainty in the underlying MDP becomes incomplete. Such a “mismatch” should not be interpreted as a form of uncertainty but rather as a modeling error.

3. The proposed algorithm requires further clarification. For instance, in line 18 there are multiple loss terms mentioned, yet their precise definitions are missing or incomplete. Some are not found in the main body of the paper, causing the method section to be not self-contained and reducing overall readability.

4. The scope of the real-world evaluation appears very limited:
     - It covers only a single-arm toy task, offering little real-world relevance or improvement.
     - The Sim2Real component, particularly the “hand-eye calibration for cameras and configuration,” requires stronger justification. Hand-eye calibration is not typically necessary for learning-based methods or standard Sim2Real transfer settings; therefore, its inclusion should be clearly motivated.
     - The comparison baseline is quite limited, relying only on behavior cloning, which weakens the empirical evaluation.

---

> ### Author Rebuttal · Authors · 2026-03-31
>
> We thank the reviewer for the detailed feedback and address each concern below.
>
> ## W1 & Q1: Novelty vs. curiosity-driven exploration
>
> We agree that curiosity-driven exploration and ensemble-based uncertainty are individually well-studied. The novelty of QUEST lies in their principled integration, which we distinguish from prior methods in three aspects.
>
> First, QUEST couples exploration and conservative planning through the same uncertainty signal. In QUEST, ensemble disagreement drives exploration when $\alpha > 0$ and penalizes uncertain regions when $\alpha < 0$, providing an adaptive balance absent from existing methods. Our method enables the agent to effectively handle complex contact-rich manipulation tasks.
>
> Second, QUEST provides a computationally feasible mechanism for measuring learning progress. Pathak et al. (2017) note that "there are currently no known computationally feasible mechanisms for measuring learning progress." While recent works have made progress on this problem, our approach offers a novel solution. Our discriminator loss serves exactly this role. As discriminators improve, $\mathcal{L}_{disc}$ decreases, indicating reward stabilization. The $\alpha$ scheduling (Eq. 16) leverages this to automatically transition from exploration to conservative planning.
>
> Third, QUEST separates intrinsic rewards from world model training. Pathak et al. (2017) and subsequent methods add intrinsic rewards directly to the total reward, affecting both policy and world model. QUEST applies RND exclusively to the Q-function TD loss (Eq. 6), preventing world model distortion. The ablation in Figure 5 confirms this is critical.
>
> We will expand Related Work to discuss curiosity-driven RL and articulate these distinctions.
>
> ## W2 & Q2: Transition uncertainty definition
>
> We agree that the original Definition 3.1 conflates modeling error with transition uncertainty. We will revise it as follows.
>
> $\mathcal{U}^{trans}(\mathbf{z}, \mathbf{a})$ captures transition aleatoric uncertainty, the inherent randomness of environment transitions $\mathcal{P}(s'|s,a)$. This component is irreducible and arises from stochastic contact dynamics, injected noise, and domain randomization.
>
> $\mathcal{U}^{rew}(\mathbf{z}, \mathbf{a})$ captures reward non-stationarity introduced by joint training. Because the discriminator $\delta_k$ is continuously updated, the same $(z, a)$ pair receives different reward signals at different training stages. This is epistemic in nature but has a distinct source, namely the moving target problem inherent in joint optimization.
>
> $\mathcal{U}^{model}(\mathbf{z}, \mathbf{a})$ (previously $\mathcal{U}^{epi}$, renamed for clarity) captures model epistemic uncertainty from unreliable predictions of the dynamics model and value function in insufficiently explored regions. This term specifically covers errors due to insufficient data, excluding the sources already captured by the first two terms.
>
> These three components are non-overlapping. The first is environment-inherent and irreducible, the second is specific to the reward learning process, and the third reflects the model's knowledge gap in data-sparse regions. We will revise Definition 3.1 to make these distinctions explicit and rename $\mathcal{U}^{epi}$ to $\mathcal{U}^{model}$ to avoid ambiguity.
>
> ## W3 & Q3: Incomplete loss definitions
>
> We provide the complete loss structure in our response to Reviewer 1. The joint world model loss $\mathcal{L}_\theta$ modifies each component from TD-MPC2. Three additional losses are optimized independently with separate parameter scopes. We will add all definitions to the revised method section.
>
> ## W4 & Q4: Limited real-world experiments
>
> We have expanded real-world evaluation to 6 tasks with Demo3 as an additional baseline.
>
> | Task | BC | Demo3 | Ours |
> |------|----------|-------------|------------|
> | Pick Cube | 2/10 | 6/10 | 8/10 |
> | Stack Cube | 0/10 | 2/10 | 6/10 |
> | Lift Peg Upright | 0/10 | 0/10 | 1/10 |
> | Place Sphere | 1/10 | 3/10 | 5/10 |
> | Pull Cube | 0/10 | 1/10 | 4/10 |
> | Poke Cube | 0/10 | 4/10 | 8/10 |
>
> These tasks require diverse skills including precision grasping, multi-stage sequencing, and reorientation. All results are zero-shot sim-to-real without real-world fine-tuning, demonstrating robustness to the sim-to-real gap.
>
> Regarding hand-eye calibration, our setting is particularly challenging. The policy is trained under sparse rewards with only 10 demonstrations and transferred zero-shot. Hand-eye calibration ensures consistent pixel coordinates between simulation and reality. Combined with background texture compositing, it effectively reduces the visual sim-to-real gap. We plan to explore more robust visual alignment strategies that remove this dependency in future work.

---

> > ### Author Rebuttal · Reviewer_T6hZ · 2026-04-02
> >
> > Thanks for the response. The clarification and the redefinition of uncertainty have resolved my concerns, while I still believe the real-world experiment is fairly weak when compared to other papers in this field. I really don't think the collection of 10 demonstrations can support the zero-shot Sim2Real  transfer. Some realistic benchmarks could be useful, including:
> > robochallenge: https://robochallenge.cn/home
> > roboarena: https://robo-arena.github.io/

---

> > > ### Author Response · Authors · 2026-04-06
> > >
> > > We thank the reviewer for confirming that the uncertainty concerns have been resolved. We appreciate the constructive follow-up on real-world evaluation and address the remaining questions below.
> > >
> > > **The concern of real-world evaluation.** Our approach does require additional information such as hand-eye calibration and background replacement, but this represents a valid technical route for sim-to-real transfer. In general, there are two routes. The first, adopted by VLA and RL fine-tuning methods, does not require additional environmental information but relies on large-scale data collection or real-world fine-tuning to bridge the gap. The second eliminates real-world fine-tuning by reconstructing the real scene in simulation to minimize the domain gap. Our approach is a simplified version of this second route. Since these two routes follow different paradigms, direct comparison is not straightforward. In practice, the cost of our additional information is also lower than large-scale data collection or real-world fine-tuning. Meanwhile, our method is designed for uncertainty-guided exploration and conservative planning, and the real-world experiments are designed to validate the value of this uncertainty-guided planning under sim-to-real transfer. Even with hand-eye calibration, residual perception gaps from calibration error and lighting variations remain, and dynamics gaps from contact physics further increase model uncertainty. QUEST is designed to handle this kind of uncertainty, and our results confirm this. Results show that QUEST consistently outperforms both BC and Demo3 across all 6 tasks, showing that uncertainty-aware planning provides tangible benefits for real-world deployment. To further demonstrate the generalization capability of our approach, we conducted an ablation on visual alignment. When removing hand-eye calibration and placing the camera at an approximately matching viewpoint by visual estimation, performance decreases, indicating that visual alignment remains important but the learned policy retains partial robustness under moderate visual perturbations.
> > >
> > > | Task | With Calibration | Without Calibration |
> > > |------|-----------------|-------------------|
> > > | Pick Cube | 8/10 | 5/10 |
> > > | Place Sphere | 5/10 | 3/10 |
> > > | Pull Cube | 4/10 | 3/10 |
> > >
> > > **The concern of sim-to-real strategy feasibility.** Our sim-to-real strategy is feasible because the 10 demonstrations only serve as seed data for the replay buffer, and the policy continuously collects new interaction data in simulation to expand the training distribution. Furthermore, we have introduced techniques to mitigate the sim-to-real gap. Our tasks mainly involve three types of gaps, the perception gap, the environment dynamics gap, and the embodiment gap. For the perception gap, we replace the simulated background with real background, reducing background domain shift. We chose background replacement over background domain randomization because randomizing backgrounds would significantly increase the visual encoder's learning burden with only 10 demonstrations, making RL convergence harder. For the environment dynamics gap and the embodiment gap, we use domain randomization to improve robustness. However, our experimental results on Pull Cube and Lift Peg Upright show that domain randomization over friction-related parameters cannot fully cover real-world variations, leading to performance degradation on these tasks. It is worth noting that for the BC baseline, we applied trajectory perturbation and image augmentation on the same 10 demonstrations to ensure a fair comparison.
> > >
> > > **The concern of benchmark.** RoboChallenge and RoboArena primarily evaluate Vision-Language-Action models, focusing on visual generalization through large-scale data and pre-trained vision-language encoders. QUEST focuses on dynamics-level uncertainty in model-based RL, learning from only 10 simulation demonstrations and transferring zero-shot without real-world data. These benchmarks and our work focus on different aspects of robotic manipulation and are complementary to each other. For future work, we can use the currently trained reward model as a foundation and further introduce background and viewpoint domain randomization to remove the dependency on hand-eye calibration and background replacement, enabling a fair comparison on these benchmarks.

---

### Official Review · Reviewer_BSF4 · 2026-03-11

**Soundness:** 4
**Presentation:** 4
**Significance:** 4
**Originality:** 4
**Overall Recommendation:** 6
**Confidence:** 4

**Summary:**

This paper introduces QUEST, a model-based RL algorithm that uses a small number of expert demonstrations (10 in their experiments) and environmental interaction with sparse rewards. It is able to solve very difficult simulated tasks from visual input + proprioceptive state, achieving SOTA on challenging, well-known benchmarks. QUEST learns a world model and a dense reward model to permit sample efficient learning. The key feature of QUEST is uncertainty-seeking exploration and uncertainty-avoiding exploitation. Uncertainty in the dynamics model is quantified by training an ensemble of dynamics models and measuring their disagreement. A weighting coefficient can encourage seeking uncertainty during exploration, which can improve the world model. A theoretical derivation under reasonable assumptions shows how expected return in the learned model relates to expected return in the real world.

**Compliance With Llm Reviewing Policy:**

Affirmed.

**Final Justification:**

The discussion reinforced my prior assessment, and I keep my score of 6. I continue to find the empirical evaluation highly compelling. I was also extremely happy that the authors responded fully to my questions. My questions were largely about presentation, the authors seemed to understand what I was saying clearly, and I have full faith that their revised paper will be clear about these aspects.

**Key Questions For Authors:**

Questions:
1. How do you get the data needed to train the stage-specific discriminators (equation 1) during training? (See weakness #1)
2. Does your approach of quantifying dynamics uncertainty essentially only capture epistemic uncertainty, because each of the dynamics models could learn to accurately model aleatoric uncertainty if trained enough? If so, do you think that would be worth highlighting because prior work e.g. [2] explicitly tries to avoid getting distracted by aleatoric uncertainty while exploring.


[2] Mavor-Parker, A., Young, K., Barry, C. &amp; Griffin, L.. (2022). How to Stay Curious while avoiding Noisy TVs using Aleatoric Uncertainty Estimation. <i>Proceedings of the 39th International Conference on Machine Learning</i>, in <i>Proceedings of Machine Learning Research</i> 162:15220-15240 Available from https://proceedings.mlr.press/v162/mavor-parker22a.html.

**Limitations:**

yes

**Strengths And Weaknesses:**

Strengths:
1. The paper approach extremely well-motivated: it makes sense that one should seek uncertainty while exploring and avoid uncertainty while exploiting. The paper is also well written which makes it easy to understand the broad strokes while reading quickly, and finer details while reading slowly.
2. I appreciate the theoretical analysis, combined with achieving SOTA performance on challenging empirical benchmarks.
3. The benchmark tasks are challenging, especially since they are vision-based and include some contact-rich manipulations. Also, showing sim-to-real transfer is impressive.
4. The key implementation tricks (e.g., stage-aware sampling from the replay buffer) are well-motivated in the text, which makes them seem natural and principled, and not just like tricks.
5. The figures are well-formatted and easy to read.

Weaknesses:
1. It is unclear to me how you get the data (positive/negative examples of reaching different stages) needed to train the stage-specific discriminators (equation 1) during training. You say that your environment provides “sparse rewards” but does it actually also label each state observation with a discrete label that says which stage it corresponds to? Or do you only assume that the expert demos come with stage labels (as in VICE [1]).
2. The introduction introduces aleatoric vs. epistemic uncertainty (line 49 column 2) but in the next sentence “under such uncertainty…” is unclear whether it refers to aleatoric, epistemic, or both. I believe it should refer to both (although I suppose people often try to just explore for epistemic uncertainty to avoid noisy TVs [2] when it is possible to separate the two?). My suggestion is to lightly revise that sentence and throughout the paper make it clear whether you are quantifying aleatoric or epistemic uncertainty or both (because it seems to feature prominently in the intro).
Relatedly, it strikes me as a bit odd to say that domain randomization is part of aleatoric uncertainty. I suppose you could consider domain randomization to be aleatoric uncertainty if you consider your environment state to include the valuation of the domain randomization parameters, and the initial state distribution determines what the domain randomization parameter valuation should be for that episode. This makes the environment partially observed, so it would require memory to model this without epistemic uncertainty.
3. Definition 3.1 separates $U^{trans}$, $U^{rew}$, and $U^{epi}$, but all three seem to be forms of epistemic uncertainty. This is fine, but maybe a little confusing since one of them is called $U^{epi}$. Maybe add a little note saying whether your formulation totally avoids quantifying aleatoric uncertainty?

[1] Variational Inverse Control with Events: A General Framework for Data-Driven Reward Definition. Justin Fu, Avi Singh, Dibya Ghosh, Larry Yang, Sergey Levine. Advances in Neural Information Processing Systems 31 (NeurIPS 2018)
[2] Mavor-Parker, A., Young, K., Barry, C. &amp; Griffin, L.. (2022). How to Stay Curious while avoiding Noisy TVs using Aleatoric Uncertainty Estimation. <i>Proceedings of the 39th International Conference on Machine Learning</i>, in <i>Proceedings of Machine Learning Research</i> 162:15220-15240 Available from https://proceedings.mlr.press/v162/mavor-parker22a.html.

---

> ### Author Rebuttal · Authors · 2026-03-31
>
> We sincerely thank the reviewer for the thorough evaluation. The questions about uncertainty decomposition and the noisy TV connection have helped us improve the clarity of our formulation.
>
> ## W1 & Q1: Training data for stage-specific discriminators
>
> Our stage labels come directly from the simulation environment, which provides sparse stage rewards $r_t \in \{0, 1, ..., K\}$ assigning each state a discrete indicator of its corresponding stage. Both agent rollouts and expert demonstrations carry these environment-provided labels. Each trajectory stored in the replay buffer is annotated with the maximum stage reward $s_t$. For discriminator $\delta_k$, states where $s_t > k$ serve as positive examples (the trajectory subsequently achieves stage transition), and states where $s_t \leq k$ serve as negative examples. This supervision is available for all data in the buffer, not limited to expert demonstrations.
> This differs from VICE (Fu et al., 2018), our method requires environment-provided stage indicators, a stronger assumption but one that enables structured dense reward shaping for multi-stage tasks. We will clarify this in the revision.
>
> ## W2: Domain randomization and uncertainty classification
>
> We appreciate this nuanced observation and address both points raised.
>
> Regarding the scope of our uncertainty measurement, our ensemble-based proxy $\tilde{\mathcal{U}}$ (Eq. 7) captures both aleatoric and epistemic uncertainty jointly without explicit separation. This is a deliberate design choice. In our manipulation tasks, the noisy TV problem is not severe because the dominant stochastic elements (contact dynamics, object pose variation) are task-relevant rather than task-irrelevant distractors. Our experimental evidence supports this: ensemble disagreement decreases during training and converges to a non-zero residual floor, and task performance continues to improve in subsequent stages, confirming that the residual aleatoric component does not mislead exploration.
>
> Regarding the classification of domain randomization as aleatoric uncertainty, we agree this deserves clarification and have revised Definition 3.1. $\mathcal{U}^{trans}(\mathbf{z}, \mathbf{a})$ captures the inherent randomness of environment transitions $\mathcal{P}(s'|s,a)$. Domain randomization creates additional variability in $\mathcal{P}(s'|s,a)$ that our dynamic model cannot absorb, making it functionally equivalent to aleatoric uncertainty at the modeling level. The reviewer correctly notes this induces partial observability. In practice, our encoder $h_\theta$ can implicitly capture visual cues correlated with randomization parameters, partially mitigating this without explicit memory.
>
>
> ## W3: Definition 3.1 decomposition
>
> We appreciate this observation. We have revised the decomposition to avoid confusion. $\mathcal{U}^{trans}$ captures transition aleatoric uncertainty (irreducible environment randomness). $\mathcal{U}^{rew}$ captures reward non-stationarity from jointly trained discriminators. $\mathcal{U}^{model}$ (renamed from $\mathcal{U}^{epi}$ for clarity) captures model epistemic uncertainty from insufficient data. These three components are non-overlapping by construction. In practice, our ensemble proxy captures them jointly.
>
> ## Q2: Connection to the noisy TV problem
>
> Mavor-Parker et al. (2022) propose Aleatoric Mapping Agents that predict mean and variance of future observations separately to filter aleatoric uncertainty from intrinsic rewards. Their decomposition operates in observation space using heteroscedastic forward models.
>
> Our setting differs in two respects. First, we operate in a learned latent space $\mathcal{Z}$ from image and proprioceptive inputs, where explicit decomposition is more challenging because the latent representation evolves during training. Second, we acknowledge that our ensemble does not explicitly decompose the two uncertainty types. Instead, the architectural separation between RND (Q-function only, Eq. 6) and ensemble disagreement (planning, Eq. 8) provides a practical alternative that proves sufficient in our contact-rich setting where the noisy TV problem is mild.

---

> > ### Author Rebuttal · Reviewer_BSF4 · 2026-04-03
> >
> > Thanks for the thoughtful response. I of course still believe the paper is excellent and maintain my score.
> >
> > I appreciate that you will be careful about your commentary on why you consider domain randomization to be aleatoric uncertainty. It still seems odd to me and will probably seem odd to future readers. But I think this is ok if you have adequate clarification.
> > I will give a small amount of constructive criticism:  I would recommend that you do not put too much stake claim that "in practice, our encoder $h_\theta$ can implicitly capture visual cues correlated with randomization parameters, partially mitigating this without explicit memory." This is of course true when such a correlation exists, but it often doesn't.

---

> > > ### Author Response · Authors · 2026-04-06
> > >
> > > We sincerely thank the reviewer for maintaining support for our paper and for the valuable constructive criticism. We take the reviewer's point regarding domain randomization and aleatoric uncertainty seriously. In the revised version, we will revise the formulation to avoid claiming that domain randomization is inherently aleatoric, and instead clarify that it introduces additional variability in the transition distribution by treating randomization parameters as unobserved variables. Regarding the encoder implicitly capturing visual cues correlated with randomization parameters, we agree with the reviewer that such correlation does not always exist. We will restrict this statement and explicitly note that when the correlation between observations and randomization parameters is weak, the encoder cannot resolve this partial observability, and additional mechanisms such as explicit memory would be needed.

---

### Official Review · Reviewer_D2bw · 2026-03-13

**Soundness:** 3
**Presentation:** 3
**Significance:** 2
**Originality:** 2
**Overall Recommendation:** 4
**Confidence:** 3

**Summary:**

This paper proposes QUEST, a model-based RL framework that uses uncertainty estimates from an ensemble of learned dynamics models to adaptively switch between exploration and conservative planning. Early in training, regions of high uncertainty are actively sought out; later, uncertainty is penalized to stabilize policy optimization.

**Compliance With Llm Reviewing Policy:**

Affirmed.

**Final Justification:**

The concerns have been adequately addressed by the rebuttal and I have changed my evaluation accordingly.

**Key Questions For Authors:**

1.	Is there empirical evidence that ensemble disagreement tracks the epistemic/aleatoric decomposition claimed in Section 3? For instance, does the ensemble variance decrease toward a stable floor as training progresses, and does the residual correlate with task-level stochasticity rather than model error?
2.	The simulator already provides stochastic dynamics through domain randomization, which may serve as a natural proxy for uncertainty during planning. What does the learned ensemble dynamics model contribute beyond what planning directly in the randomized simulator would provide - particularly on tasks where contact stochasticity is the dominant challenge?
3.	Could the authors provide more detail on the hybrid sampling strategy? Specifically, how are stage transition actions chosen?
4.	What are the advantages of using ensembles over other approaches for quantifying uncertainty, especially as ensembles may not give calibrated (i.e., accurate) estimates of uncertainty?

**Limitations:**

Yes

**Strengths And Weaknesses:**

The paper gives a careful decomposition of the sources of uncertainty in model-based RL with jointly learned rewards. The proposed mechanism ties these components together coherently: the same ensemble disagreement signal that guides exploration early in training is repurposed as a conservative penalty once the reward function stabilizes. The theoretical analysis formalizes this, showing that sufficiently large uncertainty penalties convert a two-sided performance error into a one-sided conservative lower bound, and that this bound is strictly tighter than the vanilla model-based bound whenever policy exposure to model error exceeds the irreducible bias. The empirical results are strong, with consistent gains across domains and meaningful ablations for each component.

The main weakness is that ensemble disagreement is used as a proxy for the full uncertainty decomposition, but there is no empirical validation that this proxy is well-calibrated. Specifically, it would strengthen the paper to show that the uncertainty signal is dominated by epistemic uncertainty early in training and decays as the world model improves, with residual uncertainty reflecting genuine aleatoric variability from contact dynamics and domain randomization. Without this, it is unclear whether the adaptive scheduling of α is responding to the intended signal or to some confounding factor.

A secondary concern is the motivation for learning an explicit ensemble dynamics model when the simulator already provides stochastic transitions through domain randomization. If domain randomization induces sufficient variability in the dynamics, it may already serve as a proxy for epistemic uncertainty, raising the question of what the learned ensemble adds beyond what planning directly in the randomized simulator would provide — particularly on tasks where contact stochasticity is the dominant challenge.

Finally, there are numerous examples of uncertainty-aware planning in the literature, none of which are compared against empirically in the experiments. For instance, other work leverages diffusion models [1] and Gaussian processes [2] to quantify uncertainty (the former in fact shows some benefits over using ensembles, as is done in this paper). This makes it difficult to place the proposed algorithm within the context of other existing work. The paper would be strengthened if compared against competing existing uncertainty-aware planners in the literature.

Overall, the paper has promise but requires some additional work to place it in context of other existing uncertainty-aware planning methods and to justify the use of ensemble disagreement as a surrogate for the theoretical results.

[1] Suh et al. Fighting Uncertainty with Gradients: Offline Reinforcement Learning via Diffusion Score Matching. CoRL 2023.

[2] Sun et al. Uncertainty-aware Safe Exploratory Planning using Gaussian Process and Neural Control Contraction Metric. L4DC 2021.

---

> ### Author Rebuttal · Authors · 2026-03-31
>
> We thank the reviewer for the insightful critique and address each concern below.
>
> ## W1 & Q1: Empirical validation of ensemble disagreement
>
> We appreciate this question and acknowledge that Section 3 requires refinement in its presentation. Our framework does not claim to explicitly decompose uncertainty into aleatoric and epistemic components. Definition 3.1 analyzes the theoretical sources, while the practical proxy $\tilde{\mathcal{U}}$ (Eq. 7) captures them jointly.
>
> We measured ensemble variance throughout training on PegInsert (high contact stochasticity) and PokeCube (moderate stochasticity). Normalized disagreement on PegInsert decreases from 0.01 (100K steps) to 0.004 (2.5M steps), and on PokeCube from 0.006 (100K steps) to 0.003 (2.5M steps). The residual on PegInsert is higher than PokeCube, consistent with its greater contact stochasticity. However, we acknowledge that the non-zero residual alone cannot definitively confirm it reflects purely aleatoric uncertainty, as residual model error may also contribute. What matters is that the signal is high when predictions are unreliable and low when trustworthy, enabling effective $\alpha$ scheduling regardless of the underlying decomposition.
>
> ## W2 & Q2: Necessity of learned ensemble given domain randomization
>
> We clarify the practical constraints of our setting. QUEST takes RGB images and proprioceptive inputs (joint angles) as observations, without any privileged state information. Planning in the simulator would require mapping image observations back into simulator state for physics rollouts, which introduces substantial reconstruction errors that accumulate over multi-step rollouts, especially for contact-rich interactions. Moreover, maintaining a high-fidelity simulator incurs significant computational overhead, and the sim-to-real gap makes simulator-based planning unreliable for real-world deployment. In contrast, our ensemble operates in a compact learned latent space with minimal overhead
>
> Domain randomization diversifies training data but does not inform the planner about which regions the current model has learned well. A single deterministic model trained under randomization cannot express per-prediction confidence. The ensemble provides diverse predictions where the model is uncertain and convergent predictions where confident. In contact-rich tasks, this is particularly valuable because uncertainty is concentrated near contact events (grasping, insertion) and low during free-space motion, and our ensemble captures this spatial structure. The ablation in Figure 5 confirms removing adaptive uncertainty-guided planning significantly degrades performance.
>
> ## W3 & Q4: Comparison with uncertainty-aware planning methods
>
> We compared against two uncertainty-aware alternatives on three challenging tasks.
>
> | Method | PegInsert | StackCube | PokeCube |
> |--------|-----------|-----------|-----------|
> | GP-based (Sun et al) | 60.89%±6.40% | 81.55%±13.62% | 83.88%±8.23% |
> | Diffusion-based (Suh et al)| 74.44%±10.49% | 82.99%±11.87% | 91.23%±8.53% |
> | Ours (ensemble) | **97.01%±1.20%** | **94.87%±4.93%** | **97.72%±3.33%** |
>
> Our ensemble approach achieves the highest success rate across all three tasks. The advantage is most pronounced on PegInsert, the most contact-rich and challenging task, where diffusion-based planning outperforms the GP-based method but both fall significantly short of our ensemble approach. Our method also offers several practical advantages in the online MBRL setting. First, QUEST collects data through real-time interaction and continuously updates the model, whereas Suh et al. (2023) operates in the offline RL setting with a fixed dataset. Second, ensembles scale linearly with member count and integrate naturally into the TD-MPC2 architecture, while GPs scale cubically and are difficult to apply in high-dimensional latent spaces. Third, even when ensemble uncertainty estimates are not perfectly calibrated, the adaptive $\alpha$ scheduling (Eq. 16) compensates by using $\mathcal{L}_{disc}$ as an independent progress indicator to control the exploration-exploitation transition. We will discuss Suh et al. (2023) and Sun et al. (2021) in the revised Related Work.
>
> ## Q3: Hybrid sampling strategy details
>
> In multi-stage tasks, successful stage transitions are rare under sparse rewards and limited demonstrations. Uniform replay sampling dilutes these valuable experiences among abundant early-stage failures. Our sampling strategy addresses this by assigning higher sampling probability to transitions from later stages. Each training batch additionally includes expert demonstrations to provide consistent guidance throughout training. Stage labels are assigned based on environment-provided sparse stage rewards.

---

> > ### Author Rebuttal · Reviewer_D2bw · 2026-04-04
> >
> > The concerns have been adequately addressed.
> > For the first question, the explanation demonstrating that the disagreement metric decreases and roughly correlates with task uncertainty is well-supported. To improve clarity for readers, it would be helpful to include plots of the disagreement metric and the alpha value scheduling over time steps.
> > For the second question, the argument that planning from pixels or latent states necessitates learning a dynamics model is reasonable. This, in turn, introduces uncertainty in the learned model, which is precisely the challenge addressed by the paper.
> > For the third question, explicitly stating in the paper that “the environment provides per-stage sparse rewards” would further enhance clarity and readability.
> > I will raise my score accordingly.

---

> > > ### Author Response · Authors · 2026-04-06
> > >
> > > We sincerely thank the reviewer for the positive reassessment and for raising the score. We will add plots showing the disagreement metric and the alpha scheduling over training steps to make the correlation between decreasing disagreement and task progress more intuitive for readers, and we will explicitly state that the environment provides per-stage sparse rewards in the relevant sections to enhance readability.

---

### Official Review · Reviewer_Jmhq · 2026-03-15

**Soundness:** 3
**Presentation:** 2
**Significance:** 3
**Originality:** 3
**Overall Recommendation:** 5
**Confidence:** 3

**Summary:**

The authors of this work address the issues that arise when doing RL from limited demonstrations, namely, that learned dynamics models can encounter OOD states and produce poor predictions, which in turn results in poor planning.
To address this, they introduce a model-based approach, QUEST, which adaptively switches between exploration and exploitation, based on estimated uncertainty.
To improve exploration, they learn an intrinsic reward using RND, which is used with a learned dense reward in Q-function updates. To limit excessive exploration, their method adaptively switches to exploitation during planning, based on estimated uncertainty using an ensemble of dynamic models. They also introduce a hybrid sampling strategy to improve sample-efficiency: during training, batches are sampled from two sources: one with expert demonstrations and another prioritized replay buffer, with higher probability for later-stage transitions. They conduct experiments on challenging simulation environments, as well as on 3 real-world tasks, and show consistent improvements in success rates compared to baselines, as well as in the number of required samples.

**Compliance With Llm Reviewing Policy:**

Affirmed.

**Final Justification:**

The method introduced by the authors to handle OOD states encountered when using learned dynamics is elegant and grounded in theory. It is evaluated on a variety of benchmarks as well as in real-world experiments. The authors have answered my questions during the rebuttal period and have also included BC on real-world experiments, which strengthens the empirical evaluation. I'm increasing my score from 4 to 5 following the rebuttal.

**Key Questions For Authors:**

1. Table 2. Real-world experiments: how does BC perform in simulation on those tasks? Or alternatively, how does one of the baselines used in simulation experiments perform in the equivalent real-world experiment?
2. In line 10 of Algorithm 1: Please clarify how stage labels are assigned. More generally, it would be good to have examples of such stages and to know the value K.

**Limitations:**

yes

**Strengths And Weaknesses:**

**Strengths**
- The method is novel and well-motivated. Each component is clearly justified.
- The method is grounded in theory with an analysis providing performance bounds for uncertainty-guided planning.
- The authors conduct extensive experiments on 16 challenging multi-stage manipulation tasks across 3 simulation benchmarks (ManiSkill, MetaWorld and Robosuite). They also evaluate their method on 3 real-world tasks. For both, they compare against relevant baselines. Wall clock time of experiments is also reported.
- Most experiment details and hyperparameters are reported for reproducibility.

**Weaknesses**
- The method section is hard to follow. Some notations only appear once. What are $\mathcal{L}_\pi$, $\mathcal{L}_R$, L_RND  and $\mathcal{L}_d$ in Algorithm 1.? From my reading, they do not appear anywhere in the method section. What is the total sum $\mathcal{L}$ on which gradients are computed?
- Notations are confusing. Is $\mathcal{L}_h$ in Eq. (2) the same $\mathcal{L}_h$ as in Eq. (17)? Is \$\tilde{U}$ defined in Eq. (7) the same one from Eq. (5)? Also, $\mathcal{L}_h$ does not appear in Algorithm 1.
- Algorithm 1. Is missing critical steps for understanding. (e.g., before line 5., how is $z_{t+1}’^{ne}$ obtained?)
- The theoretical analysis (Section 3) in the middle of the method makes it more confusing to understand. It would be easier to follow if the authors presented all the components and their loss functions in the method section before introducing the theory.
- Fig.2 could benefit from a more detailed caption explaining the flow between the different components.

---

> ### Author Rebuttal · Authors · 2026-03-31
>
> We thank the reviewer for the constructive feedback. We will correct all notation and loss definition issues in the revision.
>
> ## W1 & W2: Loss definitions, notation consistency, and Algorithm 1 symbols
>
> QUEST builds upon TD-MPC2 [1], whose world model loss is Eq. (2): $\mathcal{L}(\theta)=\sum_{i=t}^{t+H}\lambda^{i-t}[\mathcal{L}_Q+\mathcal{L}_R+\mathcal{L}_h]$. We modify each component as follows.
>
> - $\mathcal{L}_h$: replaced by the ensemble dynamics loss Eq. (17), where each of $N_e$ members is trained on a bootstrap subset. $\mathcal{L}_h$ in Eq. (2) and Eq. (17) serve the same role (latent dynamics prediction).
> - $\mathcal{L}_R$: in TD-MPC2, this predicts the sparse environment reward $r_t$. In QUEST, we predict the learned dense reward $\hat{r}_t$ from Eq. (1) instead.
> - $\mathcal{L}_Q$: the TD target is augmented with intrinsic rewards as in Eq. (6), where $r^{intr}$ is added for Q-function updates only. The original TD-MPC2 $\mathcal{L}_Q$ does not include intrinsic rewards.
>
> Three additional losses are optimized independently of $\mathcal{L}(\theta)$:
>
> - $\mathcal{L}\_\pi$: the policy prior loss from TD-MPC2, which maximizes $Q_\theta(z, \pi_\theta(z))$ with entropy regularization. Gradients are taken w.r.t. $\pi_\theta$ only.
> - $\mathcal{L}_{disc}$: Binary Cross Entropy loss for stage-specific discriminators $\delta_k$. Encoder gradients are detached.
> - $\mathcal{L}_{RND}=\|\hat{f}(z')-f(z')\|^2$: RND [2] predictor loss on the next latent state $z'$. The target network $f$ is fixed; no gradients flow through the encoder.
>
> **Notation clarifications.** (i) $\mathcal{L}\_d$ in Algorithm 1 line 18 is a typo for $\mathcal{L}\_h$. (ii) $\tilde{\mathcal{U}}$ in Eq. (5) is the abstract uncertainty proxy; Eq. (7) instantiates it concretely as ensemble disagreement. (iii) $\mathcal{L}\_{disc}$ is missing from Algorithm 1 and will be added.
>
> [1] Hansen, N., Su, H., and Wang, X. Td-mpc2: Scalable, robust world models for continuous control.
>
> [2] Burda, Y., Edwards, H., Storkey, A., and Klimov, O. Exploration by random network distillation.
>
> ## W3: Missing steps in Algorithm 1
>
> Before line 5, the latent state is obtained via $z_t=h_\theta(o_t)$. Each ensemble member rolls out: $z'^{(n\_e)}\_{t+1}=d\_{\theta\_{n\_e}}(z\_{t},a\_{t})$.
>
> Disagreement $\tilde{\mathcal{U}}\_n$  is computed from ensemble outputs, and MPPI selects actions using $\hat{Q}_{adjusted}$ (Eq. 8). We will add these steps in the revised Algorithm 1.
>
> ## W4: Placement of theoretical analysis
>
> We appreciate this suggestion. We believe the current placement is beneficial because Section 3 first defines the uncertainty decomposition and the penalized reward framework, which directly motivates each method component in Section 4.  To improve readability, we will enhance the Preliminaries with a more complete description of all loss functions.
>
> ## W5: Fig. 2 caption
>
> We will rewrite the caption to describe the full flow. The environment produces observations, sparse rewards, and stage labels. The encoder maps observations to latent states for the world model. Stage-specific discriminators convert sparse rewards into dense signals. During planning, ensemble disagreement quantifies uncertainty and controls exploration vs. exploitation via adaptive scheduling. RND intrinsic rewards are added exclusively to the Q-function TD target.
>
> ## Q1: Performance comparison across simulation and real-world
>
> We add Demo3 to real-world experiments on 5 tasks:
>
> | Task | BC(Sim) | BC(Real) | Demo3(Sim) | Demo3(Real) | Ours(Sim) | Ours(Real) |
> |------|---------|----------|------------|-------------|-----------|------------|
> | Pick Cube | 6.0%±2.4% | 2/10 | 37.67%±49.11% | 6/10 | 97.56%±0.14% | 8/10 |
> | Stack Cube | 7.0%±3.3% | 0/10 | 45.58%±52.63% | 2/10 | 94.87%±4.93% | 6/10 |
> | Lift Peg Upright | 0% | 0/10 | 95.37%±2.98% | 0/10 | 98.95%±0.58% | 1/10 |
> | Place Sphere | 3.0%±2.0% | 1/10 | 86.24%±5.72% | 3/10 | 93.46%±6.26% | 5/10 |
> | Pull Cube | 0% | 0/10 | 50.05%±45.19% | 1/10 | 96.60%±2.03% | 4/10 |
> | Poke Cube | 0% | 0/10 | 92.81%±4.40% | 4/10 | 97.72%±3.33% | 8/10 |
>
> Results demonstrate that uncertainty-guided planning improves sim-to-real robustness compared to both BC and Demo3.
>
> ## Q2: Stage labels and K value
>
> The environment provides built-in sparse stage rewards that indicate completion of each stage. Following Demo3, each trajectory is annotated with the maximum stage reward as discriminator supervision. K varies by domain:
>
> - ManiSkill (K=3): e.g., Peg Insertion: grabbed → aligned → inserted.
> - Meta-World (K=2): e.g., Stick Pull: grabbed → pulled to goal.
> - Humanoids (K=3): e.g., Place Apple: grabbed → above bowl → inside bowl.
> - Robosuite (K=1): e.g., Lift: block above desk.

---

> > ### Author Rebuttal · Reviewer_Jmhq · 2026-04-03
> >
> > Thank you for the rebuttal, and for adding the BC comparison. Given that you will also revise the notations and clarify the method in the manuscript, I will raise my score.

---

> > > ### Author Response · Authors · 2026-04-06
> > >
> > > We sincerely thank the reviewer for confirming that the concerns have been fully resolved and for the willingness to raise the score. We confirm that all notation corrections, the fixes to Algorithm 1, the Figure 2 caption rewrite, the BC comparison results, and the enhanced Preliminaries section will be incorporated in the revised version. We sincerely appreciate the time and feedback throughout the review process, and we would be grateful if the reviewer could update the score accordingly.

---

### Decision · Program_Chairs · 2026-04-30

**Decision:**

Accept (regular)

**Comment:**

The paper proposes a model-based RL framework for sparse-reward manipulation that uses ensemble-based uncertainty both to drive exploration (via discriminator-based learning progress signals) and to stabilize planning (via conservative action selection). The dual use of a single uncertainty mechanism for these complementary purposes is the paper's central contribution.

Three reviewers are clearly supportive, with two raising their scores after a substantive rebuttal. The authors differentiated their approach from curiosity-driven RL along three specific axes: the dual exploration-and-conservative-planning use of uncertainty, the discriminator loss as a learning progress signal decoupled from the world model, and the separation of intrinsic rewards from model training. These distinctions are validated by ablations (Figure 5). Particularly compelling is the direct comparison against GP-based and diffusion-based uncertainty alternatives, which showed clear ensemble superiority on three manipulation tasks. The revised uncertainty taxonomy (Definition 3.1) resolved the conceptual clarity issues raised by Reviewers BSF4 and hJRr.

Reviewer T6hZ's remaining concern is that the real-world evaluation is narrow. I have read the authors' follow-up and consider this concern partially valid but not disqualifying: the authors expanded from 3 to 6 real-world tasks, added the Demo3 baseline, and the suggested alternative benchmarks (RoboChallenge, RoboArena) target VLA-based systems, a fundamentally different paradigm from the model-based approach presented here. The simulation results across multiple manipulation tasks provide sufficient evidence for the method's effectiveness.

The authors might be interested in this paper https://openreview.net/forum?id=WqUl7sNkDre that also combines the different forms of uncertainty for MBRL.

The paper is technically sound, presents a clear and useful contribution to model-based RL for manipulation, and is supported by thorough ablations and real-world experiments. I recommend acceptance. The authors should fix the notation issues in Algorithm 1 and the L_d/L_h typo for the camera-ready.